# LLMSELECTOR: TOWARDS MODEL SELECTION OPTIMIZATION FOR COMPOUND AI SYSTEMS

## ABSTRACT

Compound AI systems that combine multiple LLM calls, such as Self-Refine and Multiagent-Debate, are increasingly critical to AI advancements. Perhaps surprisingly, we find empirically that choosing different models for different modules has a substantial effect on these systems' performance. Thus, we ask a core question in compound AI systems: for each LLM call or module in the system, how should one decide which LLM to use? As a first step, we formally show that the model selection problem (MSP) is computationally intractable. Next, we propose LLMSELECTOR, a principled framework that learns LLMs' strengths and weaknesses across different modules through an LLM evaluator and then performs an efficient optimization to select which models to use in any given compound system with a bounded number of modules. Our theoretical analysis gives mathematical conditions under which LLMSELECTOR only requires LLM calls scaling linearly with the number of modules and the number of LLMs to identify the optimal model selection. Extensive experiments across diverse tasks, including multimodal question answering, health knowledge comprehension, and advanced reasoning challenges, demonstrate that LLMSELECTOR achieves up to 79% gains for compound AI systems like Self-Refine, Multiagent-Debate, and Majority-Vote with frontier reasoning models including GPT-5 and Gemini 2.5 Pro. Similarly, LLMSELECTOR unlocks up to 73% performance improvements as well when using general-purpose models such as GPT-4o and Claude 3.5 Sonnet.

## 1 INTRODUCTION

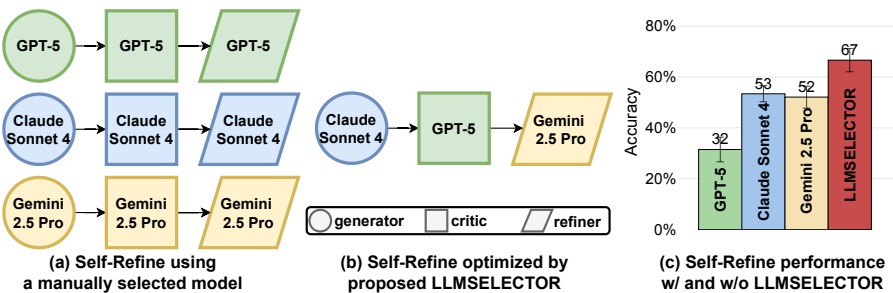

Figure 1: Demonstration of LLMSELECTOR on Self-Refine, a widely-used compound AI system. Self-Refine consists of three modules: a generator, a critic, and a refiner. (a) Vanilla Self-Refine requires users to manually select a model for all modules, limited by the model's ability to handle each module effectively. (b) Instead, our proposed LLMSELECTOR autonomously learns to select the best-suited model per module while considering interdependencies between modules for end-to-end optimization. (c) On a real-world dataset (Word Sorting), LLMSELECTOR brings large performance gains (14%) to Self-Refine over manually allocating any fixed model. Note that LLMSELECTOR is also applicable to many other compound systems such as Majority-Vote and Multiagent-Debate.

Researchers and developers are increasingly leveraging large language models (LLMs) by composing multiple LLM calls in a compound AI system to tackle complex tasks (Du et al., 2024; Zhang et al., 2024b; Madaan et al., 2023; DeepMind, 2023; Shinn et al., 2023; Renze & Guven, 2024;

Zaharia et al., 2024). For example, a common practice is to use one LLM call to generate one initial answer, one LLM call to give feedback, and one more call to refine the answer based on the feedback, known as Self-Refine (Renze & Guven, 2024; Madaan et al., 2023; Ji et al., 2023). Another example is Multiagent-Debate (Du et al., 2024; Liang et al., 2024; Khan et al., 2024), where multiple LLM calls are made to propose initial answers and then debate which ones are correct. Compared to making a single model call, significant improvements are possible because the compound systems decompose challenging tasks into simpler sub-tasks, and perform one LLM call for each sub-task.

Most existing work on compound systems focuses on optimizing prompts used in individual modules and/or module interactions, while using the same LLM for all modules (Khattab et al., 2024; Yuksekgonul et al., 2024; Wu et al., 2023; Chase et al., 2022). While this simplifies compound system design, it leaves important questions unaddressed. In particular, does using different models across modules improve a compound system's performance? Perhaps surprisingly, we find empirically that these choices have a substantial effect on quality: different models are better at different modules. Then how should one select which model to use for each module? With the growing number of LLM calls in compound systems and available LLMs, automated model selection is critical to enhance generation quality, simplify decision-making, and improve accessibility for non-experts.

We take a first step by systematically studying model selection in one of the most widely used families of systems, namely, static compound AI systems, i.e., those where the number of modules, the sequencing of module calls, and the mapping between modules and models are fixed. In this context, we find that allocating different LLMs to different modules leads to a significant increase in performance than allocating the same LLM to all modules (Figure 1). As an example, consider again the Self-Refine system (Madaan et al., 2023) consisting of three modules: a generator, a critic, and a refiner. LLM A may be better at providing feedback but worse at generating and refining answers than LLM B. In this case, allocating LLM A for the critic and LLM B for the generator and refiner is better than allocating either one to all modules.

Next, we formulate the model selection problem (MSP), i.e., identifying the best model each module should use to maximize the overall performance. MSP is challenging in principle, as it is infeasible to exhaustively search the exponentially large space of all model choices. More precisely, there are $|M|^{|V|}$ choices, where $|V|$ is the number of components, and $|M|$ is the number of models. We show that choosing the models optimally involves solving a problem that is NP-Hard.

However, in this paper we show that solving MSP is possible with much lower complexity, specifically, $O(|M| \cdot |V|)$. This leverages two key insights we make that apply to many cases: (i) the end-to-end performance can be monotonic in per-module performance, i.e., if you replace the model of a component with a better model, the end-to-end system's performance will improve, and (ii) per-module performance can be estimated accurately by an LLM evaluator. This motivates us to design LLMSELECTOR, a framework that tackles MSP efficiently for any static system with provable guarantees on performance optimality and linear computation complexity under mild assumptions. LLMSELECTOR first learns the strengths and weaknesses of each model on different modules via an LLM evaluator. Then it initializes each module with the learned best model and iteratively updates each module. This is applicable to any compound system whose number of modules is fixed. Furthermore, LLMSELECTOR incurs only limited overhead. We provide the mathematical conditions under which LLMSELECTOR finds the optimal solution for MSP with linear complexity, i.e., uses a number of LLM calls that is linear in the number of modules and models (Section 4).

We conduct systematic experiments on a diverse set of compound AI systems using frontier reasoning models (including GPT-5, Claude Sonnet 4, and Gemini 2.5 Pro) as well as general-purpose LLMs (including GPT-4o, Claude 3.5 Sonnet, and Gemini 1.5 Pro), for a range of tasks, such as multimodal question answering, health knowledge comprehension, and advanced reasoning challenges. Given a task, choosing a model carelessly easily leads to more than 50% accuracy drop than using a carefully selected model. LLMSELECTOR achieves 2%-79% performance gains compared to allocating the same LLM to all modules using reasoning models (Figure 5 in Section 5) and 4%-73% using general-purpose models (Figure 6 in Section 5). LLMSELECTOR also outperforms advanced techniques specializing in prompt optimization (Table 3 in the appendix). This further highlights the importance of model selection for compound AI systems. In short, our main contributions are:

- **Model selection problem.** We formulate the model selection problem (MSP) for compound AI systems, an increasingly important but under-explored problem. We have found

empirically that allocating different models to different modules has large performance effects (up to 100%), and show formally that optimizing MSP is NP-Hard.

- **The LLMSELECTOR framework.** To optimize MSP, we propose LLMSELECTOR, a principled framework that learns the strengths and weaknesses of each model across different modules via an LLM evaluator, and then performs an efficient optimization to select which modules to use. We give mathematical conditions under which LLMSELECTOR finds the optimal solution for MSP with linear complexity, i.e., uses a number of LLM calls that is linear in the number of modules and models.

- **LLMSELECTOR's practical effectiveness.** Through extensive experiments on practical compound systems using frontier reasoning models (such as GPT-5 and Gemini 2.5 Pro) and general-purpose LLMs (including GPT-4o, Claude 3.5 Sonnet, and Gemini 1.5 Pro), we have found that LLMSELECTOR offers substantial performance gains (2%-79%) over a range of tasks including multimodal question answering and complex reasoning challenges.

## 2 RELATED WORK

**Compound AI system optimization.** Prompt engineering and module interaction design is a central topic of compound AI system optimization. While existing work often relies on manually tuning them (DeepMind, 2023; Shinn et al., 2023; Zhou et al., 2024b; Pryzant et al., 2023; Fourney et al., 2024; Zhao et al., 2024; Lu et al., 2023; Zhao et al., 2024), recent work studies how to automate this process, such as DSPy (Khattab et al., 2024), Textgrad (Yuksekgonul et al., 2024), and Autogen (Wu et al., 2023; Zhang et al., 2024a). For example, DSPy uses Bayesian optimization to adjust prompts for all modules, while Textgrad uses textual feedback to optimize prompts for individual modules. On the other hand, our work focuses on model selection, a third axis for compound system optimization, complementary to prompt optimization and module interaction design.

**Model market utilization.** Model market utilization studies how to use all available models for downstream tasks (Lu et al., 2024a; Ramírez et al., 2024; Miao et al., 2023). Extensive work has built various techniques such as model cascade (Chen et al., 2024b), model routing (Hu et al., 2024; Stripelis et al., 2024), and mixture-of-experts (Wang et al., 2024a). While they mainly focus on *single-stage* tasks such as classification (Chen et al., 2020; Huang et al., 2025) and question answering (Chen et al., 2024b; Shekhar et al., 2024), we study model utilization for compound AI systems requiring *multiple stages*. This is much more challenging as the search space is much larger.

**Model selection.** Model selection is a critical part of classic ML and has been extensively studied in the literature (Kohavi, 1995; Akaike, 1974; Elsken et al., 2019; Raschka, 2018). While classic techniques focus on model selection for one ML task (He et al., 2021; Feurer et al., 2022; Salehin et al., 2024), compound systems involve multiple ML tasks. Thus, model selection becomes more challenging as the search space is exponentially large in the number of tasks.

**LLM-as-a-judge.** LLMs are widely used for judging complex generations, termed LLM-as-a-judge. Researchers have extensively studied how LLM judges align with human preference (Zheng et al., 2023; Shankar et al., 2024), how to improve its quality (Kim et al., 2023), how to evaluate it (Chiang et al., 2024; Chen et al., 2024a; Zeng et al., 2023), as well as many other applications (Johri et al., 2025; Dhole et al., 2024; Gu et al., 2024; Zhou et al., 2024a). In this paper, we find a novel use case of LLM-as-a-judge: evaluating module-wise performance to accelerate model selection optimization.

## 3 COMPOUND AI SYSTEMS: SCOPES AND EXAMPLES

**Static Compound AI systems.** As defined by (Zaharia et al., 2024), compound AI systems address AI tasks by synthesizing multiple components that interact with each other. Here, we denote a static compound AI system by a directed acyclic graph $G \triangleq (V, E)$, where each node $v \in V$ denotes

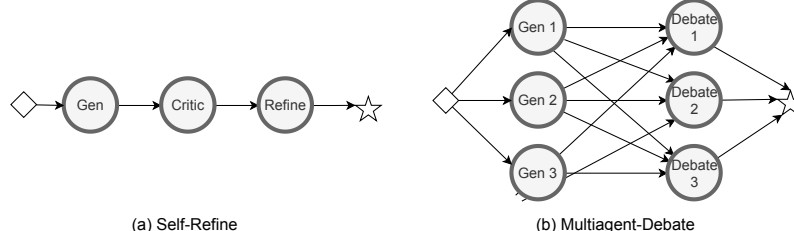

Figure 2: Examples of static compound AI systems. (a) Self-Refine system. (b) Multiagent-Debate system. The diamond and star represent the input and output modules, and the circles represent the LLM modules. Directed lines represent data flow. We omit query inputs for simplicity.

one module, and each directed edge $e \triangleq (u, v) \in E$ indicates that the output from module $u$ is sent to module $v$ as input.

**LLM modules.** An LLM module is a module that utilizes an LLM to process the inputs. It typically concatenates all inputs as a text snippet (via some prompt template), obtain an LLM's response to this snippet, and send the response as output (potentially after some postprocessing). Throughout this paper, all modules are LLM modules to simplify notations. In practice, if a module is not an LLM module, one can either merge it into an LLM module or convert it into an LLM module by conceptually "adding" an LLM to the module.

**Examples.** Consider two examples of static compound AI systems, Self-Refine and Multiagent-Debate. Self-Refine, as shown in Figure 2(a), consists of three modules: a generator, a critic, and a refiner. Given a query, the generator produces an initial answer. The critic provides feedback on the initial answer, and the refiner uses the feedback to improve the initial answer. Figure 2(b) shows the architecture of a six-module system: Multiagent-Debate. Here, three generators first give their initial answers to a question, then three debaters debate with each other based on these initial answers. Refinements and debates can be iterative, but we focus on only one iteration for simplicity.

**Notations.** Table 1 in Appendix A lists our notations. We also use $f_{i \rightarrow k}$ to indicate a function that is the same as function $f$ except that the value $i$ is mapped to the value $k$.

## 4 THE MODEL SELECTION PROBLEM: MODELING AND OPTIMIZATION

This section presents how to model and optimize model selection for static compound AI systems.

### 4.1 PROBLEM STATEMENT

Consider a static compound AI system $G = (V, E)$ and a set of LLMs $M \triangleq \{1, 2, \cdots, |M|\}$ to use. Let $\mathcal{F} : V \mapsto M$ denote all possible model allocations, each of which allocates an LLM $k \in M$ to a module $v \in V$. Given a task distribution $\mathcal{D}$, the performance of the compound AI system using the model allocation $f \in \mathcal{F}$ is $P(f) \triangleq \mathbb{E}_{z \in \mathcal{D}}[p(f, z)]$. Here, $z$ denotes a task sampled from the data distribution, and $p(f, z)$ is the performance of the compound AI system on the given task $z$ using the allocation $f$. The model selection problem is modeled as maximizing the expected performance

$$\max_{f \in \mathcal{F}} P(f) \tag{1}$$

### 4.2 THE ASSUMPTIONS

Problem 1 is challenging without any assumptions. In fact, as the search space grows exponentially in the number of modules $|V|$, we can actually show that Problem 1 is NP-Hard.

**Lemma 4.1.** *Problem 1 is NP-Hard in $|V|$ (number of modules).*

The full proof is left to Appendix B. The proof key is reducing any 3-SAT to MSP. This can be done by mapping modules to Boolean variables and model allocations to variable assignments, and

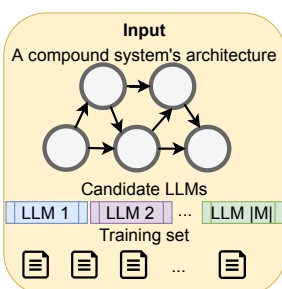 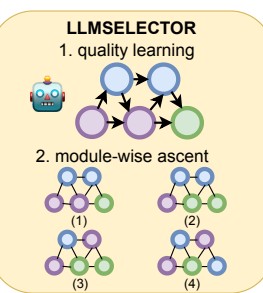 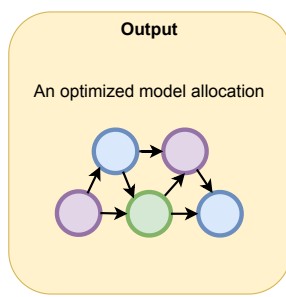

Figure 3: LLMSELECTOR workflow. LLMSELECTOR takes as input a compound AI system's architecture, a pool of candidate LLMs, and a training dataset consisting of question-answer pairs. The first step, quality learning, uses an LLM evaluator to learn each model's effectiveness on different modules. The second step, module-wise ascent, iteratively optimizes model allocation to one module while fixing other modules' allocations. This is repeated until no performance gain is possible or a training budget $B$ is reached. Finally, LLMSELECTOR returns an optimized model allocation.

a clause's satisfiability to model allocations' correctness on a query. In the following, we list our assumptions to enable tractable analysis.

**Binary performance.** For simplicity, we only consider binary performance, i.e., $p(f, z) \in \{0, 1\}$.

**Decomposition to per-module performance.** In classic computing systems such as a hardware stack, optimizing individual components (such as CPU, GPU, and memory) often leads to better overall performance. Similarly, improving individual modules' quality should also lead to better overall quality of a compound AI system. Here we assume that a compound system's performance is a monotone function of individual modules' performance. Formally, let $p_i(f, z)$ denote module $v_i$'s performance on the task $z$ using allocation $f$. Then the end-to-end performance can be decomposed as $p(f, z) = h(p_1(f, z), p_2(f, z), \cdots, p_L(f, z))$, where $h(\cdot)$ is monotonically increasing.

**Monotone module-wise performance.** The module-wise performance needs to satisfy certain properties to enable us to analyze the interplay between individual modules and the compound systems. In this paper, we focus on module-wise performance $p_i$ with the following two conditions.

- $p_i$ is *intra-monotone*: $p_i(f_{i \to k}, z) \geq p_i(f_{i \to k'}, z) \implies p_i(f'_{i \to k}, z) \geq p_i(f'_{i \to k'}, z)$. In simple terms, $p_i$ induces a "ranking" for each module: no matter how models are allocated to other modules, allocating model $k$ to a given module is always "better" than model $k'$.

- $p_i$ is *inter-monotone*: $p_i(f_{i \to k}, z) > p_i(f_{i \to k'}, z) \implies \forall j, p_j(f'_{i \to k}, z) \geq p_j(f'_{i \to k'}, z)$. In other words, if module $i$th performance is higher by replacing its allocated model from A to B, then such a replacement should not hurt other modules' performance.

*Do the assumptions always hold?* The above two conditions simplify our analysis, but they are not always satisfied in practice. In these cases, while our analysis may not hold, LLMSELECTOR is still applicable and demonstrates superior performance (as shown later in Section 5).

**Optimality Characterization.** Suppose the module-wise performance is both intra-monotone and inter-monotone. Then we are able to study the optimal allocation via the lens of module-wise performance. In particular, we first argue that it is possible to find a model allocation that maximizes the performance for each module. This is because the module-wise performance is inter-monotone: improving the model used for one module can only improve the performance for other modules. The second observation is that a module-wise optimal allocation must also be the globally optimal allocation, as the end-to-end performance is monotonic with individual module-wise performance.

### 4.3 THE LLMSELECTOR FRAMEWORK

The above analysis motivates our design of LLMSELECTOR, a principled framework for efficiently optimizing model allocation in compound AI systems.

Figure 3 gives an overview of how LLMSELECTOR works. It takes the compound AI system architecture $G$, the set of LLM $M$, a training dataset $\mathcal{D}_{\text{Tr}}$, and a training budget $B$ (i.e., number of model calls divided by $|V| \cdot |\mathcal{D}_{\text{Tr}}|$) as input, and returns an optimized model allocation $\hat{f}$ as the output. Here, each data point in the training dataset $z = (q, a) \in \mathcal{D}_{\text{Tr}}$ is a question-answer pair specifying a possible question and desired answer. LLMSELECTOR consists of two stages, namely, quality learning and module-wise descent.

**Quality learning.** In the first stage, an allocation $f^a$ is learned via an LLM evaluator, which estimates the $i$th module performance for any given module $i$, task $z$ and allocation $f$, denoted by $\hat{p}_i(f, z)$. Specifically, for a given $z$, we start with some random allocation $f^{z,0}$, and iteratively update each module with the best module-wise performance estimated by the LLM evaluator:

$$f^{z,i} \leftarrow \max_{f: \exists k, f = f_{i \to k}^{z,i-1}} \hat{p}_i(f, z), where\ i = 1, 2, \cdots, |V|. \tag{2}$$

We take the majority vote as the learned allocation, i.e., $f^a \leftarrow mode(\{f^{z,|V|}\}_{z \in \mathcal{D}_{\text{Tr}}})$.

**Module-wise ascent.** The learned allocation is not necessarily optimal because the LLM evaluator can be noisy, and thus we perform additional search based on the ground-truth overall performance. Starting with the learned allocation $f^a$, we iteratively update each module by the model with the best overall performance until budget is reached or no more improvement is possible:

$$f^i \leftarrow \max_{f: \exists k, f = f_{i' \to k}^{i-1}} \sum_{z \in \mathcal{D}_{Tr}} p(f, z), where\ f^0 = f^a, i' = i \pmod{|V|}. \tag{3}$$

The details can be found in Algorithm 1. The following result shows when LLMSELECTOR can identify the optimal allocation, and we leave the proof to Appendix B due to space limit.

**Theorem 4.2.** *Algorithm 1 always terminates. Suppose Problem 1 has a unique optimal solution, for each task $z$ in $\mathcal{D}_{\text{Tr}}$, the optimal allocation is unique, and the LLM evaluator $\hat{p}_i = p_i$. Then for some constant $c > 0$, with probability at least $1 - O(\exp(|V| \ln |M| - c|\mathcal{D}_{\text{Tr}}|))$, Algorithm 1 returns the optimal solution to Problem 1 for any training budget $B \geq |M||V|$.*

Theorem 4.2 reveals several properties of LLMSELECTOR. First, LLMSELECTOR is guaranteed to converge. Second, assuming that the LLM evaluator is perfect, a small training set is sufficient to find the optimal model allocation. Indeed, the training data size only needs to grow linearly with the number of modules and log-linearly with the number of models with high probability. Finally, the number of iterations required to find the optimal solution with high probability is linear to the number of modules, much faster than a brute-force approach.

---

**Algorithm 1:** How LLMSELECTOR works.

> **Input:** A compound AI system $G = (V, E)$, a pool of $K$ candidate LLMs, a training dataset $\mathcal{D}_{\text{Tr}}$, and a training budget $B$
> **Output:** An optimized model allocation $\hat{f}$
> **1** Choose a random $f^0 \in \mathcal{F}$ // `initialize`
> **2** Compute $f^{z,i}$ by equation (2) $\forall z \in \mathcal{D}_{\text{Tr}}, i = 1, \cdots, \min\{|V|, \lfloor \frac{B}{M} \rfloor\}$
> **3** $f^0 \leftarrow mode(\{f^{z,\min\{|V|, \lfloor \frac{B}{M} \rfloor\}}\}_{z \in \mathcal{D}_{\text{Tr}}})$ // `quality learning`
> **4** Compute $f^i$ by equation (3) $\forall i = 1, \cdots, \min\{\lfloor \frac{B}{|M|} \rfloor - |V|, 0\}\}$ // `module-wise`
>     `ascent`
> **5** return $f^i$ // `optimized model choices`

---

## 5 EXPERIMENTS

We compare the performance of LLMSELECTOR with vanilla compound AI systems using real-world LLM models in this section. Our goal is three-fold: (i) understanding when and why compound systems optimized by LLMSELECTOR outperform vanilla systems quantitatively, (ii) measuring the performance gains enabled by LLMSELECTOR across different tasks qualitatively, and (iii) validating whether LLMSELECTOR is applicable to different types of AI models.

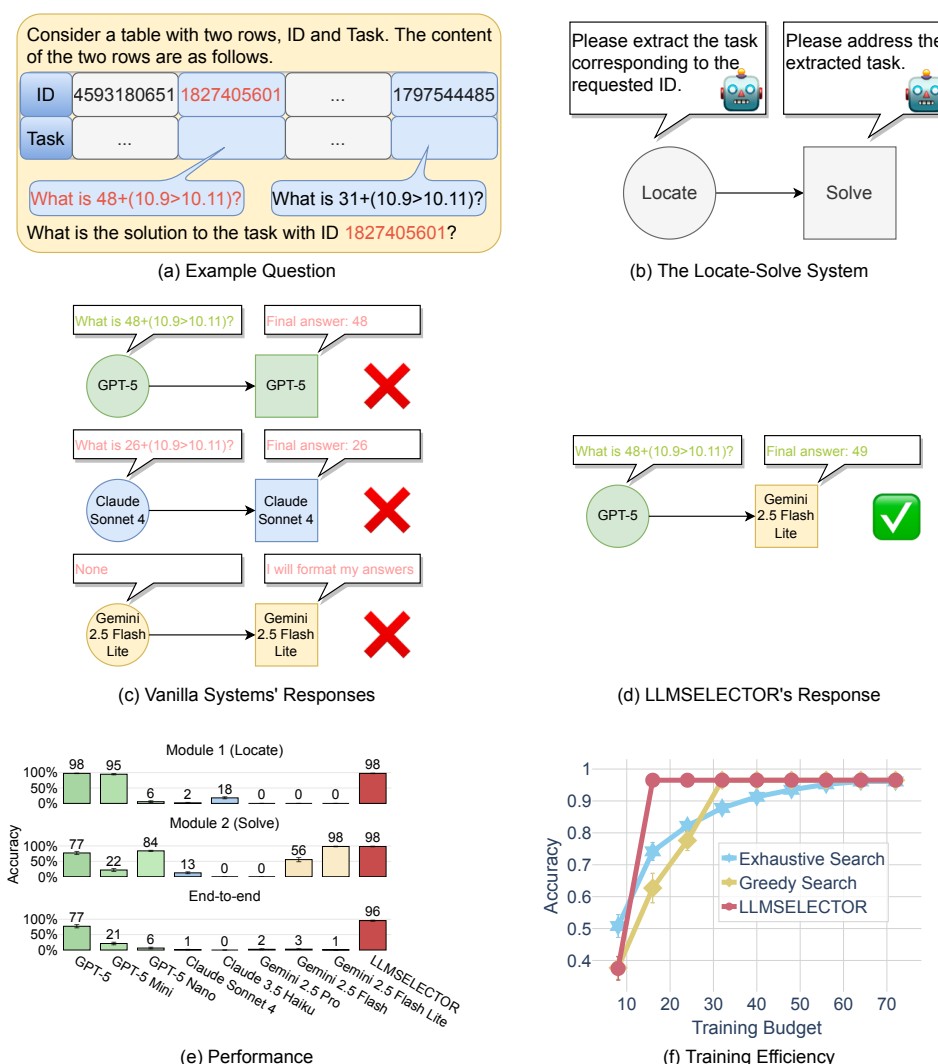

Figure 4: A case study on the TableArithmetic dataset. (a) An example question in TableArithmetic. It has a table of two rows, ID and Task, and the goal is to solve a task corresponding to a given ID. (b) We use a two-module system, Locate-Solve. The first module extracts the task, and the second module solves it. (c) Vanilla systems using a fixed model fail at the task. (d) LLMSELECTOR answers the question correctly, as it learns to use the model best-suited for each module. (e) GPT-5 performs the best for module 1 while Gemini 2.5 Flash Lite is the best for module 2. LLMSELECTOR learns to allocate GPT-5 to module 1 and Gemini 2.5 Flash Lite to module 2. Thus, its performance is substantially better than using any fixed model. (f) Compared to naive approaches, LLMSELECTOR is much more efficient. For example, it requires 75% fewer data than exhaustive search to converge.

**Experiment setups.** The main experiments are conducted with $|M| = 8$ frontier models, including GPT-5, GPT-5 Mini, GPT-5 Nano, Claude Sonnet 4, Claude 3.5 Haiku, Gemini 2.5 Pro, Gemini 2.5 Flash, and Gemini 2.5 Flash Lite. More details can be found in Appendix C.1.

## 5.1 A CASE STUDY ON TABLEARITHMETIC

We start with a case study on TableArithmetic, a synthetic dataset consisting of 100 questions. As shown in Figure 4(a), each question involves a table consisting of "ID" and "task" rows. The goal is to solve the task corresponding to a specific ID. ID is a 10-digit number, and the task is a math problem like "What is $X + (10.9 > 10.11)$?", where X is a random integer between 1 and 100. The table in each question has 200 entries in total.

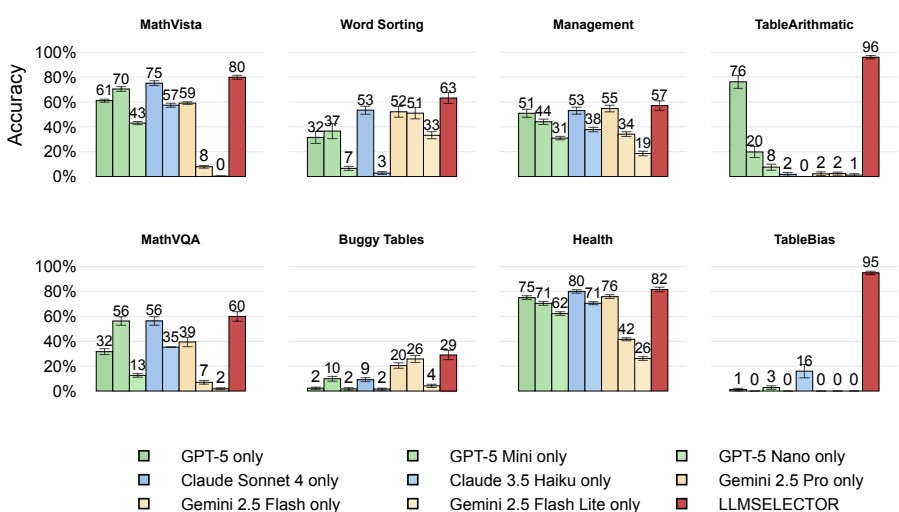

Figure 5: LLMSELECTOR's performance using frontier reasoning models including GPT-5 and Gemini 2.5 Pro. The error bar is the standard deviation over 5 runs. Overall, we have observed that LLMSELECTOR consistently offers substantial (2% to 79%) performance improvements compared to using any fixed reasoning models.

**The Locate-Solve system.** We use the Locate-Solve system using two modules for the case study. As shown in Figure 4(b), the first module, locate, extracts the task with the corresponding ID, and the second module, solve, takes the first module's output and then answers the extracted task.

**Performance gains.** We first observe that vanilla systems using one fixed model all fail to address this question. Figure 4(c) gives a few example responses. GPT-5 correctly solves the first task, but then it incorrectly believes $10.9 < 10.11$ and thus the final answer is still incorrect. Claude Sonnet 4 and Gemini 2.5 Flash Lite both fail at the first module. On the other hand, LLMSELECTOR learns to use GPT-5 for the locate module and Gemini 2.5 Flash Lite for the solve module. As shown in Figure 4(d), this leads to the correct answer 49. This is because using GPT-5 for the locate module correctly extracts the desired task, and using Gemini 2.5 Flash Lite solves the extracted task perfectly. To further understand this, Figure 4(e) shows the end-to-end as well as per-module performance of the system using each fixed model and LLMSELECTOR. Here, module 1 is considered correct if the extracted task is the desired task, and module 2 is considered correct if, given the desired task, it returns the correct final answer. One can see that GPT-5 is the best for module 1, but Gemini 2.5 Flash Lite is the best for module 2. Notably, LLMSELECTOR learns to use the best-suited model for each module without ground-truth per-module performance labels.

**Optimizer effects.** Next, we seek to understand the search efficiency of LLMSELECTOR. In particular, we compare LLMSELECTOR with two baselines: exhaustive search and greedy search. Given an LLM API budget $B$, exhaustive search samples $B$ model allocations (without replacements) from all possible allocations, and then returns the one with the highest end-to-end performance. The greedy search iteratively chooses one module and allocates to it the model with the highest end-to-end performance. As shown in Figure 4(f), we have found that LLMSELECTOR consistently outperforms these baselines. In particular, while exhaustive search needs to explore all $|M|^{|V|} = 8^2 = 64$ model allocations to ensure optimality, LLMSELECTOR needs only $|M||V| = 8 \cdot 2 = 16$ model allocations, resulting in 75% cost reduction. Interestingly, there is a tradeoff between greedy search and exhaustive search: greedy search's accuracy is higher for large budgets, while exhaustive search's performance is better for smaller budgets. One possibility is that exhaustive search imposes more diversity and inefficiency. When the budget is small, diversity implies a greater likelihood of running into a good allocation. When the budget is large, diversity is less important, and the efficiency of greedy search becomes obvious. Notably, LLMSELECTOR outperforms them for all budgets.

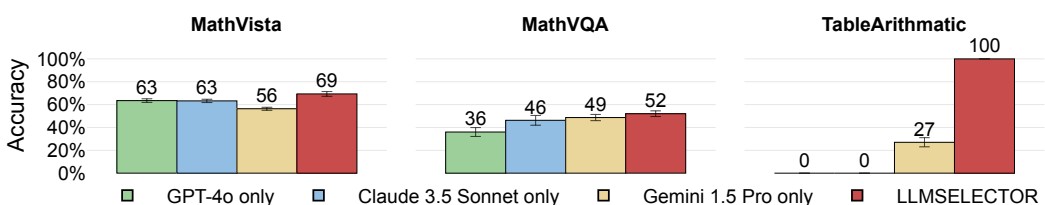

Figure 6: LLMSELECTOR's performance using general-purpose models including GPT-4o, Claude 3.5 Sonnet, and Gemini 1.5 Pro. The error bar is the standard deviation over 5 runs. Overall, we have observed that LLMSELECTOR consistently offers substantial performance improvements compared to using any fixed models.

## 5.2 MEASURING PERFORMANCE IMPROVEMENTS QUANTITATIVELY

Now we study the performance of LLMSELECTOR applied on different tasks, focusing on both frontier reasoning models and general-purpose models. For each task, we conduct five independent runs with random train-test splits, and then report the average performance as well as variance (the error bars) of using fixed models and LLMSELECTOR.

**Frontier models.** Figure 5 shows the performance of LLMSELECTOR using frontier reasoning models applied on four compound AI systems. In particular, Majority-Vote is measured on Math-Vista (Lu et al., 2024b) and MathVQA (Fu et al., 2024), Self-Refine is assessed on Word Sorting and Buggy Tables (Kazemi et al., 2025), Multiagent-Debate is evaluated on Health (Wang et al., 2024b) and Management (Du et al., 2025), and Locate-Solve is reported on TableArithmetic and TableBias. More details on the studied compound systems (such as Majority-Vote) and the datasets can be found in Appendix C.1. Overall, we observe that no LLM is universally better than all other LLMs for all tasks. For example, GPT-5 performs the best on TableArithmetic, but Claude Sonnet 4 is the best for MathVQA. Second, LLMSELECTOR offers 4%-73% performance gains consistently across different datasets and compound systems. This suggests that LLMSELECTOR is widely applicable.

**General-purpose models.** Applying LLMSELECTOR to general-purpose models also leads to substantial performance improvements. In fact, as shown in Figure 6, LLMSELECTOR brings an up to 73% performance increase when only GPT-4o, Claude 3.5 Sonnet, and Gemini 1.5 Pro are available. This suggests that LLMSELECTOR is effective across different types of LLMs. Additional experiments and details can be found in Appendix C.2 (and in particular Table 3).

## 5.3 UNDERSTANDING LLMSELECTOR'S IMPROVEMENTS QUALITATIVELY

To further understand when and why LLMSELECTOR outperforms allocating the same model to all modules, we dive into a few specific examples and compare how LLMSELECTOR's generations differ from these by allocating the same LLM. For example, we observe that LLMSELECTOR learns to allocate different LLMs to answer generators for diverse generations, but the same LLMs to debators. More details and discussions are presented in Appendix C.3 due to space limit.

## 6 CONCLUSION

The complexity of orchestrating multiple LLM calls in compound AI systems underscores the critical need for strategic model selection to optimize these systems' performance across diverse tasks. In this paper, we formalize and analyze the complexity of the model selection problem (MSP), and then propose LLMSELECTOR, a principled framework that identifies optimal model selection with provable performance and complexity guarantees, whose effectiveness has also been justified via extensive experiments on visual question answering, domain-specific knowledge comprehension, algorithmic logic challenges, and many other tasks. Extending LLMSELECTOR for user-defined inference-time budget is an interesting future direction. Discussion with AI system developers indicates that joint optimization of model selection and prompting methods is another open problem.

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
