## A  NOTATIONS

We summarize the main notations in Table 1.

Table 1: Notations.

| Symbol | Description |
| --- | --- |
| $G = (V, E)$ | A compound AI system |
| $|V|$ | Number of LLM modules |
| $M$ | The set of LLMs |
| $f : V \mapsto M$ | A model allocation |
| $z$ | One task |
| $P(f)$ | End-to-end performance |
| $p(f, z)$ | End-to-end performance on $z$ |
| $p_i(f, z)$ | $i$th module's performance on $z$ |
| $\mathcal{D}$ | The task distribution |
| $\mathcal{D}_{\text{Tr}}$ | The training dataset |

## B  MISSING PROOFS

### B.1  PROOF OF LEMMA 4.1

*Proof.* We prove that Problem equation 1 is NP-Hard via a polynomial-time reduction from the canonical NP-complete problem 3-SAT.

**Construction.**  Consider a 3-SAT instance with a CNF formula over Boolean variables $z_1, \ldots, z_m$ with clauses $C_1, \ldots, C_n$, where each clause has exactly three literals.

Construct the following compound AI system instance:

- Let $|V| = m$, with one module $v_i \in V$ corresponding to each Boolean variable $z_i$.

- Let $M = \{0, 1\}$, where model 0 represents False and model 1 represents True.

- Each allocation $f \in \mathcal{F}$ corresponds to a truth assignment to all variables.

- For each clause $C_j$, define a task $z_j \in \mathcal{D}$ whose success depends on whether clause $C_j$ is satisfied under allocation $f$.

Define:

$$p(f, z_j) \triangleq \begin{cases} 1, & \text{if } C_j \text{ is satisfied under assignment } f, \\ 0, & \text{otherwise.} \end{cases}$$

Let the data distribution $\mathcal{D} = \{z_1, \ldots, z_n\}$ be uniform over clauses. Then the expected performance becomes:

$$P(f) = \mathbb{E}_{z \in \mathcal{D}}[p(f, z)] = \frac{1}{n} \sum_{j=1}^{n} p(f, z_j),$$

which is simply the fraction of clauses satisfied by the assignment $f$.

**Reduction from 3-SAT to Problem 1.** The original 3-SAT formula is satisfiable if and only if there exists an allocation $f$ such that $P(f) = 1$. Thus, any 3-SAT instance can be solved by solving Problem 1 for the above compound AI system instance, and returning satisfiable if and only if the optimal solution is 1.

Thus, we conclude that Problem equation 1 is NP-Hard in $|V|$ (number of modules). □

## B.2 PROOF OF THEOREM 4.2

*Proof.* The first half (termination) is straightforward: Line 2 takes $\min\{|V|, \lfloor\frac{B}{M}\rfloor\}$ iterations, and line 4 takes $\min\{\lfloor\frac{B}{|M|}\rfloor - |V|, 0\}$ iterations. Thus, Algorithm 1 terminates after $\lfloor\frac{B}{|M|}\rfloor$ iterations.

Now we turn to the second half. This involves two parts. First, we show that the majority vote of all individual $f^{z,|V|}$ leads to the optimal solution to model selection on the training dataset. Next, we show that the optimal solution on the training dataset is the same as that on the data distribution with high probability. Both of them would need the following lemma.

**Lemma B.1.** *Assume $\hat{p}_i = p_i$. Then $f^{z,|V|}$ is the unique optimal allocation for the task $z$.*

*Proof.* We first note that the uniqueness of a task's optimal model allocation implies that for each module only one unique model maximizes the per-module quality. That is, for each $i$, there exists some $k$, such that for any $k' \neq k$, we have $p_i(f_{i\to k} > p_i(f_{i\to k'})$. Suppose not. Let $k^*$ be the model allocated to module $i$ by the optimal allocation. Due to the monotone assumption, $k^*$ should also maximize module $i$'s performance. Let $k'$ be another model that maximizes module $i$'s performance. By the inter-monotone assumption, switching from $k^*$ to $k'$ does not hurt any other module's performance. By the monotone assumption, $k'$ also maximizes the overall performance. A contradiction. Therefore, for each module, there is only one unique model that maximizes its performance, regardless of how other modules are allocated.

Now we can show that at the iteration $i$, allocation $f^{z,i}$ allocates the same models to the first $i$ modules as the optimal allocation. To see this, one can simply notice that the unique "best" model for each module must also be the optimal model for the end-to-end system. This is again because of the monotone assumption: otherwise, one can change the model in the optimal allocation to have better performance of one module and thus the overall system. Therefore, allocating the per-module optimal model is the same as allocating the optimal model for the entire system. Thus, at iteration $i$, allocation $f^{i,z}$ allocates the same models to the first $i$ modules as the optimal allocation. Therefore, after $|V|$ iterations, the allocation must be the unique optimal allocation for query $z$.

□

Now let us start with the first part. We first argue that the allocation learned at line 3, i.e., the majority vote of $f^{z,|V|}$ (since $B \geq |V| \cdot |M|$) over all $z$, is the optimal solution to

$$\max_f \frac{1}{|\mathcal{D}_{\text{Tr}}|} \sum_{z' \in \mathcal{D}_{\text{Tr}}} p(f, z').$$

By Lemma B.1, the optimal allocation for each query is unique. That is, $p(f, z')$ is 1 if $f = f^{z',|V|}$, and 0 otherwise. Hence, the training performance of any fixed $f$ is proportional to

$$\sum_{z' \in \mathcal{D}_{\text{Tr}}} \mathbf{1}_{f=f^{z',|V|}}$$

That is, the performance of allocation $f$ is proportional to the number of training data points whose optimal allocation is the same as $f$. Therefore, taking the majority vote of all optimal allocations is sufficient to obtain the best allocation for the training dataset.

Now we turn to the second part. By definition, $P(f) = \mathbb{E}(p(f, z)) = \mathbb{E}(\mathbf{1}_{f=f^{z,|V|}}) = \Pr[f = f^{z,|V|}]$. In words, the performance of allocation $f$ is the probability that it is the same as the optimal allocation of a query sampled from the distribution. The optimal allocation is thus $f^* = \max_{f \in \mathcal{F}} \Pr[f = f^{z,|V|}]$, where $\mathcal{F}$ is all possible allocations. The solution we obtain

at line 3, $f^a$, as shown in the first part, is the optimal solution on the training dataset, i.e., $f^a = \max_{f \in \mathcal{F}} \sum_{z \in \mathcal{D}_{\text{Tr}}} \frac{1}{|\mathcal{D}_{\text{Tr}}|} \mathbf{1}_{f=f^{z,|V|}}$. Now we can show that these two allocations are the same with high probability, by showing that for each allocation $f$ the two objectives are close to each other with high probability, and then applying the union bound.

Specifically, let $\Delta \triangleq P(f^*) = \max_{f \in F-\{f^*\}} P(f)$, i.e., $\Delta$ is the gap between the optimal allocation's performance and the second best allocation's performance. By the assumption that the optimal solution to Problem 1 is unique, we must have $\Delta > 0$. For ease of notation, let $n \triangleq |\mathcal{D}_{\text{Tr}}|$ denote the size of the training dataset, and $\hat{P}(f) \triangleq \sum_{z \in \mathcal{D}_{\text{Tr}}} \frac{1}{|\mathcal{D}_{\text{Tr}}|} \mathbf{1}_{f=f^{z,|V|}}$.

For any given allocation $f$, by Hoeffding bound,

$$\mathbb{P}\left( \left| P(f) - \hat{P}(f) \right| \geq \epsilon \right) \leq 2 \exp\left( -2n\epsilon^2 \right)$$

Set $\epsilon = \Delta/2$. This implies that with probability at least $1 - 2\exp\left(-n\Delta^2/2\right)$, $\left| P(f) - \hat{P}(f) \right| < \Delta/2$. By union bound, for all allocation $f$, with probability at least $1 - 2|\mathcal{F}|\exp\left(-n\Delta^2/2\right)$, $\left| P(f) - \hat{P}(f) \right| < \Delta/2$ holds for all $f$. Now, this suggests that for any $f \neq f^*$, we have $\hat{P}(f^*) - P(f^*) > -\Delta/2$, and $\hat{P}(f) - P(f) < \Delta/2$. Therefore, we can have

$$\begin{aligned}
\hat{P}(f^*) - \hat{P}(f) =& \hat{P}(f^*) - P(f^*) - (\hat{P}(f) - P(f)) + (P(f^*) - P(f)) \\
> & -\Delta/2 - \Delta/2 + (P(f^*) - P(f)) \\
=& P(f^*) - P(f) - \Delta \geq 0
\end{aligned}$$

where the last $\geq$ is be definition of $\Delta$. That is to say, the performance of $f^*$ on the training dataset is higher than that of any other allocation with high probability. Hence, the allocation that maximizes the performance on the training dataset must be the same allocation that maximizes the performance on the data distribution, with probability at least $1 - 2|F|\exp\left(-n\Delta^2/2\right)$. Recall that there are $|M|^{|V|}$ many possible allocations and thus $|\mathcal{F}| = |M|^{|V|}$ and also that $n = \mathcal{D}_{\text{Tr}}$ by definition. Thus, with probability at least $1 - 2\exp\left(|V|\ln|M| - |\mathcal{D}_{\text{Tr}}|\Delta^2/2\right)$, the obtained allocation by Algorithm 1 is the optimal allocation, which finishes the proof.

$\square$

## C  EXPERIMENT SUPPLEMENTS

Here we present more details on the experiments, including full experiment setups, more quantitative results, and additional qualitative analyses.

### C.1  EXPERIMENT SETUPS

#### C.1.1  COMPOUND AI SYSTEMS

In this paper, we focus on four compound AI systems, Locate-Solve, Self-Refine, Multiagent-Debate, and Majority-Vote. Their architectures are shown in Figure 7. Locate-Solve designed for TableArithmetic and TableBias consists of two modules: the first module extracts the task associated with an ID from an input table, and the second module returns the answer to the extracted task. Self-Refine (Madaan et al., 2023) has a generator, a critic, and a refiner. The generator gives an initial answer to a question, the critic gives feedback to this answer, and the refiner uses the feedback to refine the original answer. Multiagent-Debate (Du et al., 2024) involves two types of modules: answer generators and debaters. The answer generators offer initial answers to a question. The debaters take the initial answers and then debate which one is correct. In this paper, we focus on a six-module Multiagent-Debate: three modules are answer generators, and the other three are the debaters. Another simple yet widely-used compound AI system is Majority-Vote. Majority-Vote contains a few generator modules, where each module generates an independent response to the user query. Then Majority-Vote takes the majority vote over all responses as the final output. In our experiments, we use Majority-Vote with 7 generator modules.

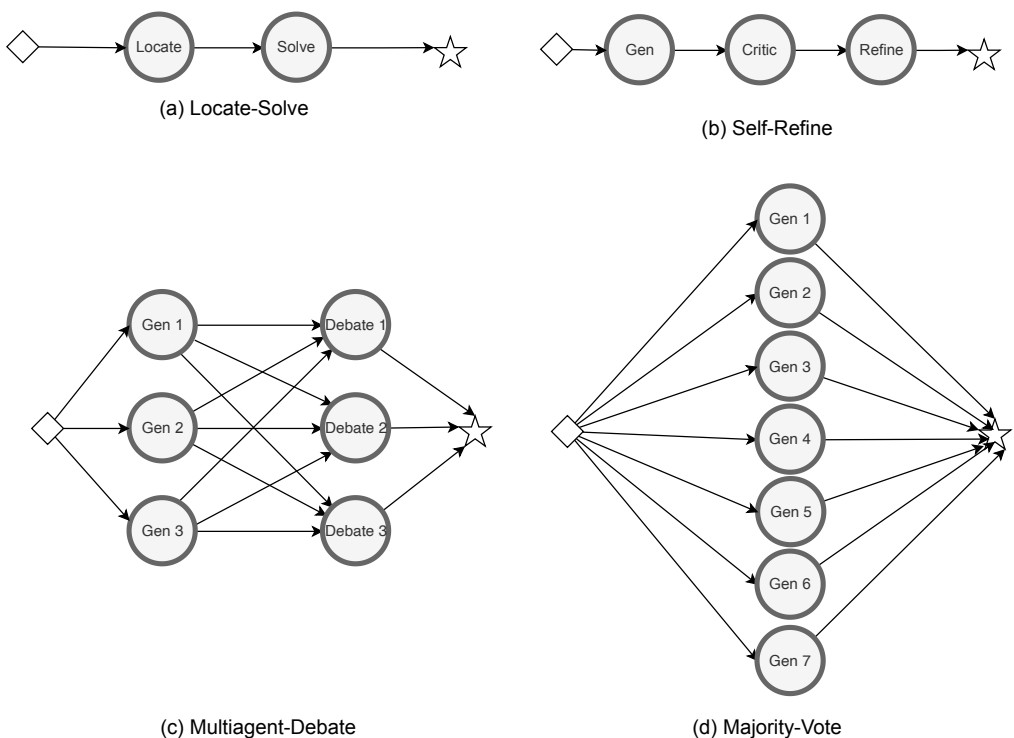

Figure 7: The architectures of the compound AI systems studied in the experiments. (a) Locate-Solve using two modules. (b) Self-Refine using three modules. (c) Multiagent-Debate that involves six modules in total. (d) Majority-Vote that leverages seven modules.

### C.1.2 DATASETS AND EVALUATION METRICS

Now we provide details of all datasets used in this paper.

**MathVista.** MathVista (Lu et al., 2024b) is a visual question answering dataset focused on challenging math problems. Each question contains one image and a math problem, and the goal is to select one out of a few given options as the answer. Since the ground-truth answers are not publicly available on the full test partition, we turn to use the entire testmini partition. We also remove a few questions from the dataset which probe OpenAI reasoning model policy violation or contain extremely large images ($> 3000 \times 5000$ pixels). This results in a total of 991 questions. This is under a CC-BY-SA-4.0 license.

**MathVQA.** MathVQA is the math subset of the OCRBenchV2 (Fu et al., 2024). We again remove the tasks that probe OpenAI reasoning model policy violation or contain extremely large images ($> 3000 \times 5000$ pixels). This results in 299 math questions involving images. Compared to other multimodal tasks, images in MathVQA often contain intense texts and thus require the model to extract the texts carefully. This dataset is under a MIT license.

**Word Sorting.** Word Sorting corresponds to the word sorting task originally from the BBEH dataset (Kazemi et al., 2025). Word Sorting contains two subtasks. One is to sort words in a modified alphabetical order. The other one is to identify errors in a given sorted traces. In total Word Sorting contains 200 questions. We use the fuzzy match provided in BBEH as the evaluation metric. It is under an Apache-2.0 license.

**Buggy Tables.** Buggy Tables is another task in the BBEH dataset (Kazemi et al., 2025). Here, each question is to reconstruct a large table given the description of a bug, and then perform some queries on the table. There are in total 200 questions. Again, we use the fuzzy match provided in BBEH as the evaluation metric. It is under an Apache-2.0 license.

**Health.** Health corresponds to the Health discipline in the MMLU-Pro dataset (Wang et al., 2024b). Each question in Health asks a health-related multiple-choice question. The model needs to choose from a large number of options (A, B, C, D, E, F, etc). We sample 800 questions for our experiments. This is under an Apache-2.0 license.

**Management.** Management corresponds to the management category in SuperGPQA (Du et al., 2025). Similar to Health, each query is a multiple-choice question. It contains 500 questions in total. This is under a ODC Attribution License.

**LiveCodeBench.** LiveCodeBench (Jain et al., 2024) is a benchmark for code understanding. We use the code execution task in LiveCodeBench [1]. It contains 479 questions in total. Each question contains a program and an input. The goal is to predict the output of the program. Note that this is a generative task, as the output space of a given program is unbounded. We use the exact match to measure the performance of a compound system's generation. This dataset is under the MIT License.

**CommonGenHard.** CommonGenHard (Madaan et al., 2023) is a constrained generation dataset consisting of 200 questions. Each question gives 20-30 concepts, and the goal is to generate a coherent paragraph that uses all the provided concepts. Since all LLMs used in our evaluation generate coherent texts, we focus on evaluating the quality of whether all concepts are included. That is, the quality is 1 if all concepts are contained in the generated paragraph, and 0 if any concept is missing. This dataset is under the Apache-2.0 License.

**SimpleQA.** SimpleQA (Wei et al., 2024) contains 4326 short, fact-seeking questions. Example questions include "Who received the IEEE Frank Rosenblatt Award in 2010" and "What is the first and last name of the woman whom the British linguist Bernard Comrie married in 1985". While seemingly simple, LLMs actually struggle to answer them correctly. We use the exact match to measure the generation quality of a compound system. This dataset is under the Apache-2.0 License.

**FEVER.** FEVER (Thorne et al., 2018) is a fact-verification dataset. We use the v2.0 variant consisting of 2384 questions [2]. Each question contains a claim, and the task is to classify the claim as one of NOT ENOUGH INFO, SUPPORTS, and REFUTES. Again, we use exact match as the accuracy metric. This dataset is under the Creative Commons Attribution Share Alike 3.0 License.

**TableArithmetic.** TableArithmetic is a synthetic dataset used to understand the locate-solve system's performance. It contains 100 questions. Each question consists of a table of "ID" and "task" rows, and the goal is to solve the task associated with a specific ID. Each row contains 100 entries. Each question has the form of "What is X+(10.9¿10.11)?", where X is a randomly generated integer.

**TableBias.** TableArithmetic is another synthetic dataset. It contains 100 questions. Each question consists of a table of "ID" and "task" rows, and the goal is to solve the task associated with a specific ID. Here, each table contains 80 entries. Each question has the form of "The surgeon, who is the boy's father, says I cannot operate on this boy, he is my son. Who is the doctor to the boy? (Ax) Father (Bx) Mother", where again x is a randomly generated integer.

### C.1.3 LLM ENDPOINTS AND PROVIDERS

We give the details of all models used in our experiments in Table 2, including their API endpoints and model providers for reproducibility purposes.

### C.2 QUANTITATIVE RESULTS

### C.2.1 FULL EVALUATIONS WITH LEGACY MODELS

Now we present the performance of LLMSELECTOR on practical compound AI systems using legacy models. We compare LLMSELECTOR with using any fixed model for all modules and

---

[1] https://huggingface.co/datasets/livecodebench/execution-v2
[2] https://huggingface.co/datasets/fever/fever

Table 2: Overview of all LLMs used in this papers. We use $|M| = 8$ models for the main experiments using frontier models, and $|M| = 10$ models for the additional experiments using legacy models. The model endpoints and providers are detailed here for reproducibility.

| Type | Model | API Endpoint | Provider |
|------|-------|-------------|----------|
| Frontier | GPT-5 | gpt-5-2025-08-07 | OpenAI |
| Frontier | GPT-5 Mini | gpt-5-mini-2025-08-07 | OpenAI |
| Frontier | GPT-5 Nano | gpt-5-nano-2025-08-07 | OpenAI |
| Frontier | Claude Sonnet 4 | claude-sonnet-4-20250514 | Anthropic |
| Frontier | Claude 3.5 Haiku | claude-3-5-haiku-20241022 | Anthropic |
| Frontier | Gemini 2.5 Pro | gemini-2.5-pro | Google |
| Frontier | Gemini 2.5 Flash | gemini-2.5-flash | Google |
| Frontier | Gemini 2.5 Flash Lite | gemini-2.5-flash-lite | Google |
| Legacy | GPT-4o | gpt-4o-2024-05-13 | OpenAI |
| Legacy | GPT-4o Mini | gpt-4o-mini-2024-07-18 | OpenAI |
| Legacy | GPT-4 Turbo | gpt-4-turbo-2024-04-09 | OpenAI |
| Legacy | Claude 3.5 Sonnet | claude-3-5-sonnet-20240620 | Anthropic |
| Legacy | Claude 3.5 Haiku | claude-3-haiku-20240307 | Anthropic |
| Legacy | Gemini 1.5 Pro | gemini-1.5-pro | Google |
| Legacy | Gemini 1.5 Flash | gemini-1.5-flash | Google |
| Legacy | Llama 3.1 405B | meta-llama/Meta-Llama-3.1-405B-Instruct-Turbo | Together AI |
| Legacy | Llama 3.1 70B | meta-llama/Meta-Llama-3.1-70B-Instruct-Turbo | Together AI |
| Legacy | Qwen 2.5 72B | Qwen/Qwen2.5-72B-Instruct-Turbo | Together AI |

Table 3: Performance of LLMSELECTOR and other approaches for optimizing compound AI systems. We focus on three common systems (Self-Refine, Multiagent-Debate, and Locate-Solve) each of which is evaluated on two tasks. The performance gain is the absolute improvement by LLMSE-LECTOR against the best of allocating any fixed (same) model to all modules (with underlines). We also compare LLMSELECTOR with the MIPROv2 optimizer implemented in DSPy (using GPT-4o as the LLM). We set max_bootstrapped_demos=2, max_labeled_demos=2, and all other parameters as default for MIPROv2. We also box the second-best result for each dataset. Overall, LLMSELEC-TOR achieves 4%-73% accuracy gains over allocating any fixed model to all modules. Interestingly, LLMSELECTOR also outperforms MIPROv2, which specializes in prompt optimization.

| Method | Compound AI System | | | | | |
|--------|--------------------|--|--|--|--|--|
| | Self-Refine | | Multiagent-Debate | | Locate-Solve | |
| | LiveCodeBench | CommonGenHard | SimpleQA | FEVER | TableArith | TableBias |
| GPT-4o | 85% | 39% | 20% | 64% | 0% | 0% |
| GPT-4 Turbo | 82% | 41% | 16% | 65% | 5% | 0% |
| GPT-4o mini | 71% | 9% | 5% | 62% | 1% | 0% |
| Claude 3.5 Sonnet | 90% | 62% | 20% | 61% | 0% | 0% |
| Claude 3.5 Haiku | 46% | 17% | 8% | 58% | 1% | 43% |
| Gemini 1.5 Pro | 87% | 39% | 16% | 60% | 27% | 0% |
| Gemini 1.5 Flash | 80% | 13% | 5% | 38% | 8% | 2% |
| Llama 3.1 405B | 81% | 77% | 21% | 66% | 0% | 0% |
| Llama 3.1 70B | 63% | 69% | 12% | 7% | 0% | 50% |
| Qwen 2.5 72B | 80% | 26% | 5% | 48% | 1% | 0% |
| DSPy MIPROv2 | 87% | 71% | 22% | 68% | 0% | 0% |
| LLMSELECTOR | **94%** | **87%** | **27%** | **70%** | **100%** | **100%** |
| Gains | 4% | 10% | 6% | 4% | 73% | 56% |

DSPy (Khattab et al., 2024), an open-source library specialized for prompt optimization in compound systems. For DSPy, we use the optimizer MIPROv2, which searches for best prompts using Bayesian optimization. We use GPT-4o as the backbone LLM, and set max_bootstrapped_demos=2, max_labeled_demos=2, and all other parameters as default for MIPROv2.

Table 3 summarizes the quantitative results using general-purpose models. First, we observe that no LLM is universally better than all other LLMs for all tasks. For example, Gemini-1.5 Pro performs the best on TableArthmetic, but GPT-4o is the best for FEVER. Second, LLMSELECTOR offers 4%-73% performance gains compared to the best baselines. Interestingly, LLMSELECTOR also outperforms the DSPy MIPROv2 which optimizes the prompt. This is again because different models have their own strengths and weaknesses, and prompting alone is not adequate to turn an LLM's weakness into its strength.

### C.2.2 LLM Evaluators: Prompts and Ablation Studies

**Prompt for the LLM evaluator.** We give the LLM evaluator prompt template in the following box. The LLM evaluator takes the module index $i$ and the compound AI system's description (including the description of each module, and how modules connect to each other) as input, and follows this prompt to evaluate module $i$'s performance. As we focus on binary performance, the performance is either high (1) or low (0).

---

**LLM evaluator prompt**

You are an error diagnosis expert for compound AI systems. Below is the description of a compound AI system consisting of multiple modules, a query, the generations from each module of the compound AI system, the final output, and the desired answer. Assume that the desired answer is 100% correct. If the final output matches the correct answer, generate 'error: 0'. Otherwise, analyze whether module i leads to the mistake. If so, generate 'error: 1'. Otherwise, generate 'error: 0'. Think step by step.
[Compound AI system]:
[query]:
[module 0 output]:
[module 1 output]:
...:
[module $|V|$ output]:
[final output]:
[desired answer]:
[your analysis]:

---

**Model for the LLM evaluator.** Unless specified, we use GPT-5 as the LLM evaluator in LLM-SELECTOR for all experiments with frontier models, and Gemini 1.5 Pro for all experiments with general-purpose models. We give a detailed ablation study on the model effects as follows.

**Effects of LLM evaluator with general-purpose models.** Here we study how different LLM evaluators affect LLMSELECTOR's performance. In particular, we use three different LLM evaluators, namely, Gemini 1.5 Pro, GPT-4o, and Claude 3.5 Sonnet, and measure their evaluation accuracy as well as end-to-end system performance, i.e., how the learned Locate-Solve system performs on the testing dataset. As shown in Table 4, we first observe that the evaluation accuracy does vary across different evaluators. Gemini 1.5 Pro's evaluation accuracy is the highest (85%), while GPT-4o's accuracy is only 68.4%. On the other hand, we observe that the end-to-end performance by using any of these LLM evaluators is impressive. This suggests that the LLM evaluators do not need to be perfect to obtain a high-quality model allocation. Finally, we note that the evaluator accuracy has an impact on "convergence rate", i.e., the training budget required to reach the optimal model allocation. When Gemini 1.5 Pro is the evaluator, budget=2 (the number of modules) is sufficient. This is because the LLM evaluator is near-optimal and thus the allocation anchoring is sufficient to find the optimal allocation, matching our theoretical analysis as well. When the LLM evaluator is noisy (such as GPT-4o), additional budget is needed for the module-wise ascent.

**Effects of LLM evaluator with frontier models.** Here we extend our study on LLM evaluator's effects to frontier models measured on diverse compound systems and tasks. More precisely, we focus on Majority-Vote on the MathVista dataset, Self-Refine on the Word Sorting dataset, and

Table 4: Effects of different LLM evaluators for the Locate-Solve system on TableArithmetic using all 10 models. All LLM evaluators lead to a high end-to-end performance. Gemini 1.5 Pro's evaluation accuracy is the highest and thus requires the smallest training budget to find the optimal model allocation. On the other hand, using GPT-4o as the evaluator leads to a lower evaluation accuracy and thus requires more training budget.

| LLM Evaluator | Evaluator Accuracy (%) | Required Budget | End-to-End Accuracy (%) |
|---|---|---|---|
| GPT-4o | 68.4 | 40 | 100 |
| Claude 3.5 Sonnet | 70.1 | 40 | 100 |
| Gemini 1.5 Pro | 85.0 | 20 | 100 |

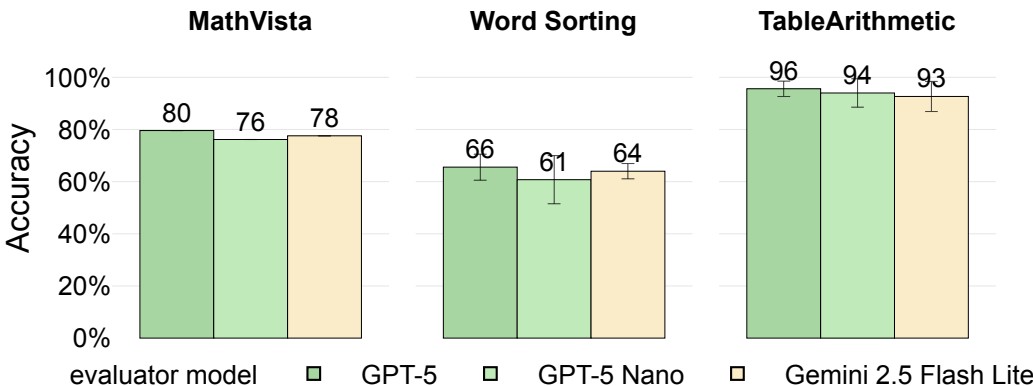

Figure 8: The effects of LLM evaluator on LLMSELECTOR applied on different compound systems. Here, we measure the performance of LLMSELECTOR using different LLM evaluators, namely, GPT-5, GPT-5 Nano, and Gemini 2.5 Flash Lite. Overall, we observe that a stronger LLM evaluator offers better performance. Indeed, GPT-5 as the LLM evaluator always leads to the best performance, while using Gemini 2.5 Flash Lite or GPT-5 Nano as the evaluator produces slightly lower accuracy. Notably, regardless of which model is used as the LLM evaluator, LLMSELECTOR's performance is consistently better than using any fixed model for the given compound AI system, suggesting its robustness.

Locate-Solve on the TableArithmetic dataset. For each case, we optimize their model allocation with LLMSELECTOR, using three different models as the LLM evaluator, namely, GPT-5, GPT-5 Nano, and Gemini 2.5 Flash Lite. The accuracy is averaged over 5 runs, and the error bar indicates one standard deviation. Overall, we observe that, as expected, strong LLM evaluators lead to better overall accuracy. As shown in Figure 8, GPT-5 as the LLM evaluator outperforms GPT-5 Nano and Gemini 2.5 Flash Lite as the evaluator. Second, across our experiments, LLMSELECTOR's performance is consistently better than allocating any fixed model to the given compound system, no matter which evaluator is used in the quality learning phase. For example, LLMSELECTOR using GPT-5 Nano as the LLM evaluator enables Self-Refine to reach 61% on the Word Sorting datasets, while allocating a fixed model to Self-Refine leads to no higher than 53%. This suggests that LLMSELECTOR is robust to the LLM evaluator's model. .

### C.2.3 ACCURACY AND COST TRADEOFFS

Now we study the accuracy and cost tradeoffs achieved by LLMSELECTOR on a range of compound AI systems and datasets. Specifically, we measure the accuracy and training budget on four cases: Majority-Vote applied on the MathVista dataset, Self-Refine applied on the Word Sorting dataset, Multiagent-Debate applied on the Health dataset, and Locate-Solve applied on the TableArithmetic dataset. Accuracy is measured by answer correctness, and cost is measured by the number of model calls, normalized by the number of training data points.

We have several interesting observations. First, LLMSELECTOR enables flexible system performance by allowing users to specify the training budgets. For example, suppose one with a small

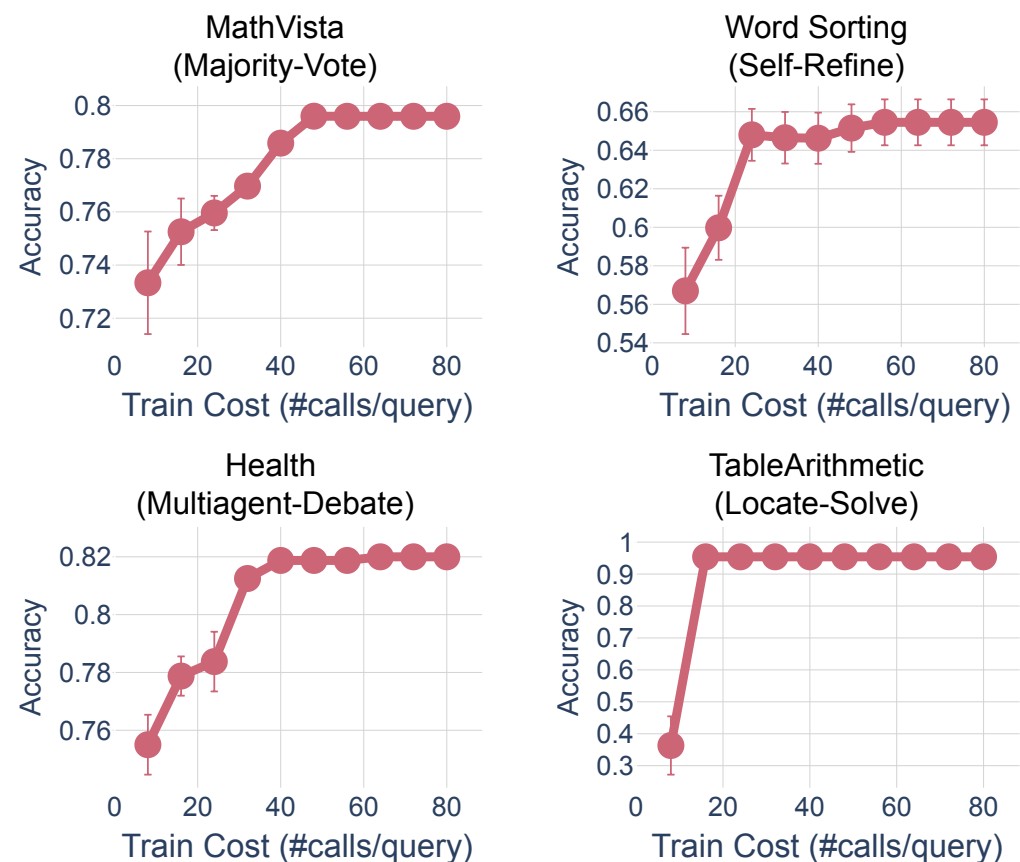

Figure 9: Accuracy and training cost tradeoffs achieved by LLMSELECTOR across different datasets and compound AI systems. Here cost (i.e., training budget) is measured by the number of model calls, normalized by the training data sizes. Overall, we observe that a few dozen model calls/query are often sufficient for LLMSELECTOR to converge. For example, training LLMSELECTOR to reach the highest accuracy of Multiagent-Debate on the Health dataset only requires 40 model calls per training query. Furthermore, LLMSELECTOR offers flexible training costs by allowing user-defined budgets. For example, much fewer model calls are needed if the goal is to surpass using a fixed model: LLMSELECTOR with 8 units of training cost has unlocked 56% accuracy on Word Sorting, while the same compound system using the best fixed model only reaches 53%.

training budget simply wants to outperform using the best fixed model to Self-Refine, whose accuracy is 53% on the Word Sorting dataset. As shown in Figure 9, 10 model calls per training query is sufficient for this user. Second, convergence of LLMSELECTOR often requires only dozens of model calls/query. For example, as shown in Figure 9, 40 model calls per training query are sufficient to find the optimal model allocation of Multiagent-Debate on the Health dataset. This roughly justifies our complexity analysis even though the monotonicity assumptions may not hold in these real-world datasets. Thirdly, we note that the training cost is a one-time cost. For large-scale applications where the compound systems are used on hundreds of thousands of inference queries, the training cost on a relatively small training dataset (containing a few hundreds to thousands) is often negligible.

### C.2.4 QUALITY LEARNING ABLATION

Next, we give an ablation study on the quality learning component. In particular, we compare the performance of LLMSELECTOR and LLMSELECTOR without the quality learning component, with a focus on the Locate-Solve system applied on the TableArithmetic dataset.

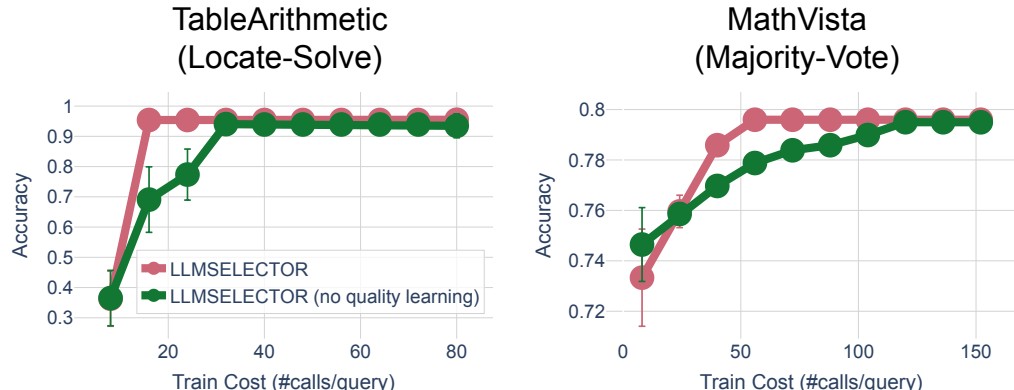

Figure 10: Quality learning's effects on LLMSELECTOR. Here, we measure both accuracy and cost of LLMSELECTOR and LLMSELECTOR without the quality learning applied on (i) the Locate-Solve system with the TableArithmetic dataset, and (ii) the Majority-Vote system with the MathVista dataset. Overall, quality learning enables a fast convergence rate. In fact, 50 model calls can converge with quality learning on the Majority-Vote system, while more than 100 are needed without quality learning. Second, we note that the converged accuracy is also slightly lower without quality learning. This implies that quality learning is a critical component in LLMSELECTOR.

We first note that the quality learning component accelerates the convergence rate of LLMSELECTOR. For example, as shown in Figure 10, LLMSELECTOR without quality learning takes 30 training units to converge on the TableArithmetic dataset, while LLMSELECTOR with quality learning only requires 16 training units, leading to a 47% training cost reduction. Second, the converged accuracy of LLMSELECTOR without quality learning is also slightly lower than that with quality learning. This is probably because no quality learning renders LLMSELECTOR converge to a locally optimal solution more often.

## C.3 QUALITATIVE ANALYSES

To better understand when and why LLMSELECTOR can outperform allocating any fixed LLM to all modules, we give qualitative examples for Self-Refine and Multiagent-Debate here.

The first example shown in Figure 11(a) is a task from the SimpleQA dataset. Allocating GPT-4o to all modules leads to an incorrect answer as seen in Figure 11(b). This is because the GPT-4o generators always return 8 as the initial answers, and the debaters fail to identify this mistake. On the other hand, LLMSELECTOR, as demonstrated in Figure 11(c), learns to allocate GPT-4o, Llama 3.1 405B, and Gemini 1.5 Pro for the three answer generators separately, and use GPT-4o for the three debaters. In this case, the three generators give completely different answers: 8, 3, and -18. Interestingly, the GPT-4o debaters reaches the consensus of 3, which is indeed the correct answer.

Another example from the LiveCodeBench dataset is shown in Figure 11(d). Here we focus on the Self-Refine system which contains three modules (a generator, a critic, and a refiner). Recall that allocating Claude 3.5 Sonnet to all modules is better than allocating any other fixed LLMs, as shown in Table 3. However, this leads to an incorrect answer for this example, as shown in Figure 11(e). This is because Claude 3.5 Sonnet as the critic mistakenly tags its initial generation as correct. On the other hand, LLMSELECTOR learns to allocate Claude 3.5 Sonnet for the generator and the refiner, but GPT-4o for the critic. As shown in Figure 11(f), this leads to a correct response to the task. This is because GPT-4o is better than Claude 3.5 Sonnet as a critic for LiveCodeBench tasks.

To sum up, LLMSELECTOR performs better than allocating any fixed models to all modules, because it identifies the strengths and weaknesses of different models across modules, and then allocate to each module the model that best fits it.

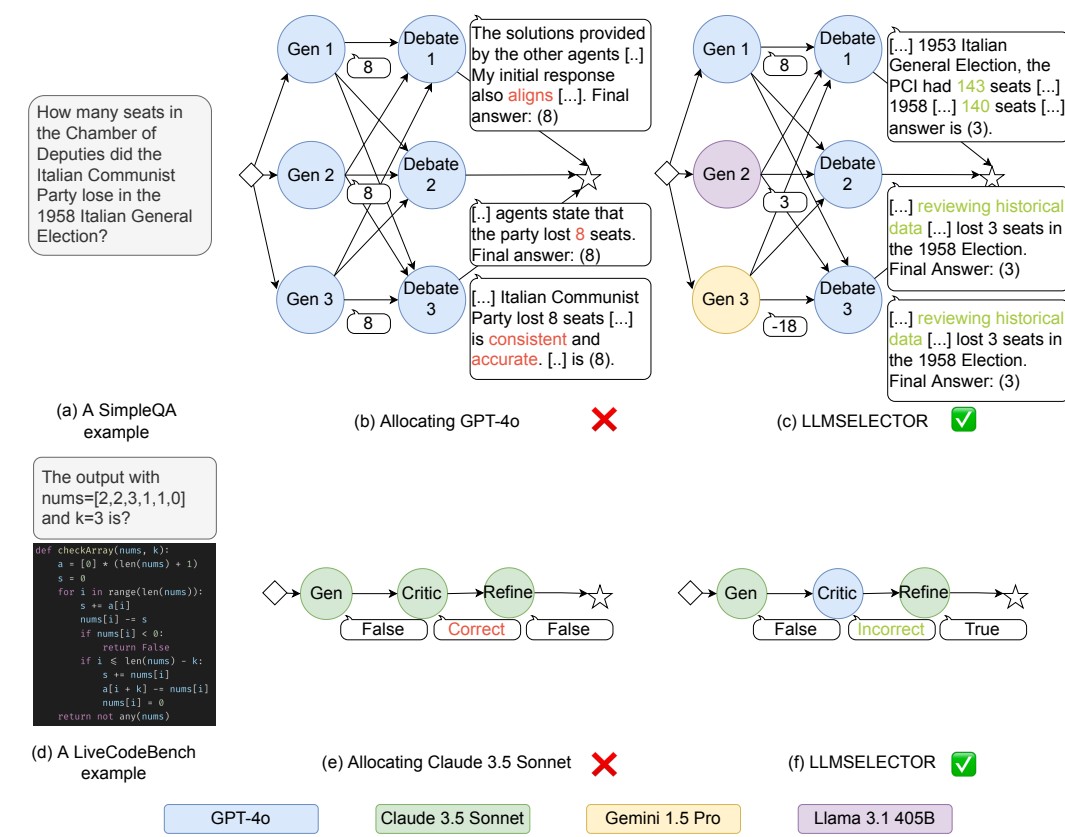

Figure 11: An Illustrative example of applying LLMSELECTOR on Multiagent-Debate on SimpleQA. (a) the task. (b) allocating GPT-4o to all modules. GPT-4o as the generator consistently generates the incorrect answer 8; thus, the debaters fail to identify this issue and lead to an incorrect answer. (c) model allocation learned by LLMSELECTOR. By allocating GPT-4o, Gemini 1.5 Pro, and LLama 3.1 405B to the three generators separately, LLMSELECTOR enables a diverse set of initial answers, and thus the debaters recognize the correct answer.

## C.4 ADDITIONAL DISCUSSION

Here we provide additional discussions and analysis of LLMSELECTOR's behaviors.

**Sensitivity to assumption violations** .The monotonicity assumptions serve two purposes: (i) enabling clean forms of performance guarantees and (ii) formalizing intuitive properties observed on compound AI systems. Algorithm 1, on the other hand, does not require these assumptions. Empirically, Algorithm 1 consistently leads to notable performance gains on real-world tasks when the monotonicity assumptions do not necessarily hold. E.g., as shown in Figure 5, applying LLMS-ELECTOR to Self-Refine leads to 10% accuracy gains, even though the monotonicity assumptions may not hold for Self-Refine. This suggests that LLMSelector is robust to imperfect monotonicity and that the assumptions, while useful for analysis, are not critical for strong empirical performance.

**Applicability to dynamic compound systems.** The goal of this paper is to lay the foundation of LLM selection in compound AI systems, an under-explored area in compound system optimization. As a first step, we focus on static systems, simple, clean, and widely-used scenarios where the technical depth and empirical benefits of LLM selection have already emerged. Extending this to dynamic compound AI systems would be an exciting future direction.

**Behavior heterogeneity across models.** Behavior and performance heterogeneity across different models have been observed in different scenarios (Chen et al., 2024b; Chiang et al., 2024; Hu et al.,

2024). Our case study in Section 5 also verifies that GPT-5 is best at locating a query, while Gemini 2.5 Flash Lite performs perfectly on solving a located query in TableArithmetic. While identifying the precise root cause is challenging, there are many possible reasons for such heterogeneity, such as differences in training data, model architectures, model sizes, pre-training/post-training algorithms, and optimization hyperparameters. How to exploit such differences will remain a long-standing challenge.

**Complexity when number of models is extremely large.** To ensure accurate model selection, a complexity linear in the number of models is unavoidable, as each model has to be evaluated at least once. When the number of available models is extremely large and a linear complexity becomes too high, the complexity can be reduced by filtering the available models first before applying LLM-SELECTOR. For example, one may only keep the models which secure the top few positions on well-known benchmarks or the most recent models. Another option is randomly choosing a few models, evaluating their performance on the training data, and only keeping those with highest performance. We leave a more principled framework efficiently handling extremely large number of models as a future direction.

## D  LIMITATIONS AND BROADER IMPACTS

LLMSELECTOR focuses on optimizing compound AI systems with a bounded number of LLM calls, and it remains open how to select models for compound AI systems with a dynamic or unlimited number of LLM calls. Based on discussions with practitioners, it is also an interesting question to jointly optimize model selection and prompting methods.

Compound AI systems that make multiple LLM calls are a rapidly growing industry with broad economic and societal impact. The large increase in available LLMs makes it inevitable to select which LLMs to use for these systems. LLMSELECTOR offers an off-the-shelf framework to automate model selection in compound AI systems. This substantially relieves users from tedious and challenging system configuration overhead. It also makes compound AI systems more accessible to more users, especially those without professional skills and knowledge in LLMs. LLMSELECTOR can optimize a compound system over any given set of LLMs, enhancing the robustness and availability of compound AI systems—even in the face of cloud outages or individual model failures—thereby supporting more reliable AI services in critical applications. We will release the code and data to stimulate more research and positive societal impacts.