# OpenReview forum: "LLMSELECTOR: Towards Model Selection Optimization for Compound AI Systems"
_ICLR.cc/2026/Conference — Submitted to ICLR 2026_

### Official Review · Reviewer_T1GQ · 2025-10-15

**Soundness:** 2
**Presentation:** 3
**Contribution:** 3
**Rating:** 6
**Confidence:** 4

**Summary:**

This paper addresses the Model Selection Problem in compound AI systems, which involve multiple LLM calls or modules. The authors observe that selecting different LLMs for different modules significantly impacts overall system performance. The MSP is formally shown to be NP-Hard due to its exponentially large search space. To tackle this, the paper proposes LLMSELECTOR, a framework that efficiently learns each model's strengths per module via an LLM evaluator and performs optimized model selection. Theoretical guarantees are provided under which LLMSELECTOR finds the optimal selection with linear complexity. Extensive experiments demonstrate performance gains of up to 79% on various tasks using both frontier and general-purpose LLMs.

**Strengths:**

1. The paper identifies and addresses the under-explored but critical problem of model selection in compound AI systems, establishing a new research direction complementary to prompt optimization and module interaction design.
2. The paper offers a solid theoretical foundation, including the NP-Hardness proof of MSP and mathematical conditions under which LLMSELECTOR achieves linear complexity and optimality.
3. Extensive experiments on multiple tasks, systems, and model types robustly validate the method's effectiveness and generality.
4. The paper demonstrates that LLMSELECTOR requires up to 75% fewer LLM calls than exhaustive search, making it highly efficient and scalable for potential real-world applications.

**Weaknesses:**

1. I think the expression in Figure 1 is not clear enough. It should more explicitly demonstrate, in a Self-Refine process, a comparison between two approaches: one using manually selected models and the other dynamically learning and autonomously selecting models, rather than presenting fixed examples. This initially may mislead readers to question, for example, why vanilla self-refine consistently selects one type of model while LLMSELECTOR chooses three different models. It would be better to even show all three mentioned tasks (Self-Refine, Multiagent-Debate, Majority-Vote) to make a full comparison and provide a whole picture for the readers.

2. In Section 1, the authors state that different models are better at different modules. However, what is the basis for this conclusion? What underlying reasons allow different models to excel at solving tasks in different modules? Since the method in this paper is built upon this assertion, I believe a more in-depth analysis and explanation are necessary.

3. The MODELSELECTOR method proposed in the paper may have limitations in practical scenarios. This method primarily targets AI systems with fixed modules. In practical applications, however, a highly autonomous intelligent agent system might dynamically plan its modules based on different inputs, thereby generating a non-fixed multi-module process. Additionally, the modules may not necessarily execute sequentially but could operate in parallel. In such cases, is this method applicable? What challenges remain to be addressed?

4. The complexity of the MODELSELECTOR method proposed in the paper is linear. However, the number of models available for selection in reality remains enormous (considering open-source model hubs such as Hugging Face or ModelScope). The experiments in the paper only considered eight general-purpose LLMs. If this method is extended to open-source model platforms, would it still be applicable? Would the complexity be too high? What technical improvements would still be needed?

5. The paper provides an explanation of quality learning, but it lacks a detailed implementation process on how to enable an LLM to learn to evaluate allocation performance, making it difficult for readers to understand how this stage is completed.

6. The author presents the performance of various tasks across eight different datasets, but there are significant differences in performance across different tasks and datasets. For example, the performance advantage in the Locate-Solve task is much greater than in the Multiagent-Debate task. What is the reason for this phenomenon? What types of tasks benefit more from model selection? This requires further analysis.

**Questions:**

See the second, the third, the fourth, and the sixth points in the weaknesses.

---

> ### Author Response · Authors · 2025-11-23
> **Thank you very much for supporting our paper!**
>
> Thank you very much for supporting our paper! We address your questions as follows.
>
> ***[Figure 1 Improvement]***: We thank the reviewer for this constructive suggestion. We have updated Figure 1 accordingly. In particular, we revise both the figure caption and subtitles to highlight that the comparison is between using a manually selected model and an automatically optimized model selection. Furthermore, we add one sentence at the end to reminder the readers that LLMSELECTOR is applicable to many other compound AI systems including Multiagent-Debate and Majority-Vote.
>
> ***[Why different models behave differently]***: Thank you for raising this important point. Our statement is based on prior empirical evidence and well-documented differences in LLM capabilities. For example, in a recent study comparing Claude Sonnet and Codex on coding tasks (https://www.linkedin.com/posts/edwinzchen_claude-sonnet-45-dropped-2-weeks-back-and-activity-7384006598920339457-l3RV?utm_source=share&utm_medium=member_desktop&rcm=ACoAABOLzmoBD1LxRnLim2e9onKGAjhvzqtiDwY), one has found that “about half the tasks each model failed were passed by the other”. A recent TMLR paper (https://openreview.net/pdf?id=cSimKw5p6R) has shown that LLama 3 (70B) and GPT-4 Turbo may excel at different tasks  (shown in their Figure 1 (c)). Our case study also verifies that GPT-5 is best at locating a query, while Gemini 2.5 Flash Lite performs perfectly on solving a located query in TableArithmetic (see Figure 4(e)). While identifying the precise root cause is challenging, there are many possible reasons for such heterogeneity, such as differences in training data, model architectures, model sizes, pre-training/post-training algorithms, and optimization hyperparameters. The discussion has also been added to Appendix C.4.
>
>
> ***[Applicability to dynamic systems]***: Thank you for the insightful question. Our method is designed for compound AI systems with a fixed set of modules, which is a widely adopted structure in current practice (e.g., Self-Refine and Majority-Vote are used in many real-world cases). The objective of this paper is to lay the foundation for optimizing model selection within such systems, where module roles and capability requirements are well defined. We agree that future autonomous agents may dynamically generate, reorder, or parallelize modules. Extending model selection to these dynamic pipelines introduces additional challenges—such as modeling a potentially unbounded module graph, handling branching/parallel execution, and making context-dependent, runtime model choices. While these settings lie beyond the scope of our current work, our formulation naturally extends to more general graph-structured or adaptive pipelines, and we view this as an important direction for future research. We have added a discussion on the dynamic systems in Appendix C.4.

---

> > ### Author Response · Authors · 2025-11-23
> >
> > ***[Complexity of LLMSELECTOR when the number of models is large]***: To ensure accurate model selection, a complexity linear in the number of models is unavoidable, as each model has to be evaluated at least once. When the number of available models is extremely large and a linear complexity becomes too high, the complexity can be reduced by filtering the available models first before applying LLMSELECTOR. For example, one may only keep the models that secure the top few positions on well-known benchmarks or the most recent models. Another option is randomly choosing a few models, evaluating their performance on the training data, and only keeping those with the highest performance. We leave a more principled framework handling an extremely large number of models as a future direction. We have added a detailed discussion in Appendix C.4.
> >
> >
> > ***[Implementation details of the LLM evaluator]***: The implementation of LLM evaluator is realized via the prompt template provided in Appendix C.2.2, which guides the LLM to evaluate allocation performance given the input query, the compound system architecture, and the desired end-to-end response to the input query. For experiments with frontier models, we use GPT-5 as the LLM evaluator, and for general-purpose models, we use Gemini 1.5 Pro as the LLM evaluator. We have also conducted an extensive ablation study on the LLM evaluator, as detailed in Appendix C.2.2 and Figure 8.
> >
> >
> > ***[Explanation of the performance gaps]***: A plausible explanation is that each module in a compound system may require multiple skills, and different models vary in their proficiency across these skills. Tasks such as Self-Refine tend to have modules where certain models strongly excel at specific skills, leading to large compounded improvements when selecting models appropriately. In contrast, tasks like Multiagent-Debate involve modules that require overlapping or entangled skills, where multiple models perform similarly, and the marginal benefit of swapping models is smaller. More broadly, performance gains of model selection depend on three factors: (i) how heterogeneous the skill requirements of each module are, (ii) how substantial the performance variance across models is, and (iii) how the end-to-end performance is affected by individual modules’ performance.

---

### Official Review · Reviewer_A9Bg · 2025-10-30

**Soundness:** 2
**Presentation:** 2
**Contribution:** 2
**Rating:** 4
**Confidence:** 3

**Summary:**

This paper tackles the model selection problem (MSP) in compound AI systems composed of multiple LLM modules.

However, the biggest weakness of this article is its lack of comparison with state-of-the-art (SOTA) multi-agent systems. If the authors could provide relevant model comparisons, I would consider raising my score.

**Strengths:**

Visualizations and tables are clear, providing both quantitative performance gains and qualitative insights.

The modular ascent/design is somehow sound.

**Weaknesses:**

1. The article does not specify the training and inference costs. Following the three-stage approach described in the article to determine the optimal model combination step-by-step would theoretically result in extremely high costs.

2. There is a lack of comparison with state-of-the-art MAS systems, and all the models used are closed-source. The authors should consider adding open-source models to facilitate the reproduction of experimental results.

3. The proof of NP-hardness and several key algorithmic details are deferred to the Appendix but are critical for assessing validity. While their correctness does not need to be independently checked, the lack of a summary or sketch in the main text undermines transparency.

**Questions:**

1. How sensitive is LLMSELECTOR to the reliability/noise of LLM-based estimators? Are there any empirical studies on its robustness or its ablation effect on estimator accuracy?

2. Could the authors specify which article TableArithmetic comes from, or how their own benchmark was constructed, and how they ensured fairness?

3. Why were all recent research results directly related to multi-agent systems and model selection not included in the experimental comparison? If the comparison is only against a baseline model, it is inherently fair, since your single response requires multiple LLMs for correction and refinement. To demonstrate the superiority of the method, the authors must supplement this with performance comparisons against state-of-the-art MAS/or LLM-as-judge systems on publicly available datasets.

---

> ### Author Response · Authors · 2025-11-23
> **Thank you for your time and feedback**
>
> Thank you for your time and feedback. We address your questions below.
>
> ***[Optimization cost]***: The proposed LLMSELECTOR allows users to specify the training budget as an input to Algorithm 1 (page 6). Hence, what matters is not cost alone, but the accuracy and cost tradeoffs. In fact, the accuracy-cost tradeoffs have been measured in our case study. In particular, the red line in Figure 4(f) corresponds to the accuracy and training cost of LLMSELECTOR on TableArithmetic. For better coverage, we conduct additional experiments to measure the tradeoffs between accuracy and optimization cost on several different tasks and compound systems, as shown in Figure 9 in Appendix C.2.3. Overall, we observe that dozens of model calls/training data point are often sufficient for LLMSELECTOR to converge. For example, LLMSELECTOR’s accuracy on Self-Refine converges after 50 calls, and the number for the Multi-agent Debate system is 40. Furthermore, only a small number of model calls are needed for LLMSELECTOR to surpass using any fixed model for a given compound system. For example, 8 model calls/query have boosted the performance to 56% on Self-Refine, while Self-Refine with the best fixed model (Claude Sonnet 4) reaches only 53%. We have added the full results and analysis in Appendix C.2.3.
>
> ***[Reproduction of experimental results]***: As proponents of reproducible research, we totally agree with the reviewer on the importance of reproducing the experimental results in our paper. To facilitate this, we will release the full artifacts associated with our paper, including code, data, evaluation scripts, and all detailed logs. These logs will make sure our results are fully reproducible without any additional requirements (such as API keys).
>
>
> ***[The proof sketch of the NP-hardness]***: Thank you for your suggestions. The proof key is reducing any 3-SAT to MSP. This can be done by mapping modules to Boolean variables and model allocations to variable assignments, and a clause's satisfiability to model allocations' correctness on a query. We have added the proof sketch to the main paper (page 4 after the NP-hardness lemma).
>
> ***[LLMSELECTOR’s reliability of the LLM estimator]***: An ablation study on different LLM evaluators’ effects on the Locate-Solve system using general-purpose models is presented in Appendix C2.2. For better coverage, we have conducted additional experiments to study the effects of LLM evaluators when applying LLMSelector to various systems and tasks. In particular, we use GPT-5, GPT-5 Nano, and Gemini 2.5 Flash Lite as the LLM evaluator separately, and evaluate the end-to-end performance of LLMSELECTOR. As shown in Figure 8, we observe that even weak LLM evaluators enable LLMSelector to outperform vanilla compound systems. For example, LLMSELECTOR using GPT-5 Nano as the LLM evaluator enables Self-Refine to attain 61% on the Word Sorting dataset, while Self-Refine using the best fixed model only achieves 53%. More details and analysis can be found in Appendix C.2.2.
>
> ***[TableArithmetic source]***: Appendix C.1.2 details the datasets used in this paper. In particular, TableArithmetic (line 896) is “a synthetic dataset used to understand the locate-solve system’s performance”. It contains 100 questions, and each question contains a table of ID and query. We will release this dataset for reproducibility and accessibility.
>
> ***[Lack of MAS baselines]***: Thank you for the valuable comment. We believe there may be a misunderstanding of our setup. Our method is not compared against single LLMs. All systems in our experiments—both baseline and ours—use the same multi-agent/compound architecture with the same modules. The only difference is how the model for each module is chosen: the baseline uses fixed/manual assignments, while our method optimizes these assignments. Therefore, the comparison isolates the effect of model selection and is fair.

---

### Official Review · Reviewer_ujw1 · 2025-10-30

**Soundness:** 2
**Presentation:** 2
**Contribution:** 3
**Rating:** 4
**Confidence:** 3

**Summary:**

This paper argues that for compound AI systems (like Self-Refine ), using different LLMs for different modules is better than using one model for everything. They propose LLMSELECTOR, a two-stage method: 1) Use an "LLM evaluator" to find a decent starting allocation ("quality learning") , and 2) Run a greedy search ("module-wise ascent") 。

**Strengths:**

1.As systems get more complex, we need to know where to use our best (and most expensive) models.

2.Clear Motivation:** The case study in Figure 4 (TableArithmetic) is excellent. It shows a task where all single models fail (0-2% accuracy), but their method gets 96% by picking the right model for each step (e.g., GPT-5 for "Locate", Gemini 2.5 Flash Lite for "Solve").

**Weaknesses:**

---

1.   The method seems to be just a combination of two known things: "LLM-as-a-judge" (Stage 1) and standard "coordinate ascent" or greedy search (Stage 2's "module-wise ascent"). This isn't a new algorithm.

2.  The linear-time guarantee (Thm. 4.2) relies on an inter-monotone assumption—that improving one module never hurts another—which rarely holds in practice (e.g., a stronger generator can make the critic’s job harder).

3. Stage 1 ("quality learning") is the most "novel" part. The paper *never* says which LLM they used as the evaluator, how they prompted it, or how robust it is. The theory assumes a *perfect* evaluator, which is impossible.

4. The large reported gains (0→96%) come mainly from synthetic datasets like TableArithmetic, while on standard benchmarks such as MathVQA the improvement is minor (56→60%).



**This gap makes the results look dataset-specific and raises doubts about the method’s real-world relevance.**

**Questions:**

---

Q1.   To justify Stage 1 (quality learning), can you please provide a key ablation: comparing LLMSELECTOR to "just running Stage 2 (module-wise ascent) from *multiple random* starting allocations"? This is crucial to prove Stage 1 is more than just a good initialization.





Q2.   Please specify the *exact model* and *the prompts* used for the "LLM evaluator" in Stage 1. This is essential for reproducibility. (b) Can you provide a sensitivity analysis? How much does final performance drop if you use a weaker model (e.g., Claude 3.5 Haiku) as the evaluator?



Q3.  How do you explain the huge performance gap between synthetic tasks (e.g., TableArithmetic, 0% $\rightarrow$ 96%) and standard benchmarks (e.g., MathVQA, 56% $\rightarrow$ 60%)? Does this not imply the method's utility is limited to tasks with extreme, clear modular specialization?




Q4. Your theory (Thm 4.2) relies on the "inter-monotone" assumption, which you admit "is not always satisfied". Can you provide *empirical evidence* (e.g., from your Self-Refine experiments) showing *how often* this assumption is violated in practice? How badly is convergence affected when it is?



Q5. Stage 2 (module-wise ascent) requires many end-to-end system evaluations. Can you report the *total cost* (e.g., total API calls) of LLMSELECTOR in practice? The Figure 4(f) case study is small ($|V|=2, |M|=8$). How does this cost scale when $|V|$ is larger (e.g., in Multiagent-Debate, $|V|=6$)?

---

> ### Author Response · Authors · 2025-11-23
> **Thank you for your time and feedback**
>
> Thank you for your time and feedback. We address your questions as follows.
>
> ***[Ablation on Stage 1 vs Stage 2]***: We conduct an ablation comparing LLMSELECTOR with Stage 2 alone (i.e., no quality learning). As shown in Figure 10 in Appendix C.4 on page 22, quality learning accelerates the convergence rate of LLMSELECTOR. For example, when applied to the Majority-Vote system on the MathVista dataset, quality learning reduces the number of model calls required to converge from more than 100 to 50, resulting in more than 50% training cost reduction. Furthermore, LLMSELECTOR with quality learning also enjoys a slightly higher accuracy. This suggests that quality learning is indeed a critical component to learn a high-quality LLM allocation. Detailed results and analysis are documented in Appendix C.4.
>
> ***[LLM evaluator details and sensitivity analysis]***: Thank you for pointing this out. We kindly report that Appendix C.2.2 has included the requested prompts and sensitivity analysis. In particular, it gives the prompt (line 987-1002) and discusses a sensitivity analysis shown in Table 4. We use GPT-5 as the LLM evaluator for frontier models and Gemini 1.5 Pro for the general-purpose models. For better coverage, we further conduct additional sensitivity experiments with frontier models. As shown in Figure 8, we observe that even weak LLM evaluators enable LLMSelector to outperform vanilla compound systems with a fixed model. For example, LLMSELECTOR using GPT-5 Nano as the LLM evaluator enables the Self-Refine system to reach 61% accuracy on Word Sorting, while Self-Refine with the best fixed model only leads to an accuracy of 53%. More details and analysis can be found in the updated Appendix C.2.2.
>
> ***[Performance gap between synthetic and standard benchmarks]***: The performance gap is expected, as the synthetic tasks have been designed to have a clear modular structure, maximizing the benefits of specialization. Real benchmarks like MathVQA require multiple skills, contain annotation noise, and cannot be decomposed into perfectly separable subtasks. Thus, even an optimal selector would achieve a smaller improvement ceiling. What matters is that LLMSELECTOR consistently captures any specialization that does exist: it achieves near-optimal gains when specialization is strong and yields solid improvements on messy real-world tasks. Furthermore, as the number of LLMs and compound AI systems rapidly grows, manual model selection is becoming impractical. Automating this process is therefore valuable in its own right, even when performance gains are modest. Hence, instead of being limited to “extreme, clear modular specialization”, the proposed method enables automated model selection and non-negligible performance gains for a wide range of real-world tasks.
>
> ***[Practical implication of the monotonicity assumptions]***: The monotonicity assumptions are likely not satisfied for real-world systems like Self-Refine. Consider, for example, that GPT-5 as the critic might be the best to carefully judge Gemini 2.5 Pro as the generator, but may be poor at analyzing its own generation due to biases. Rigorously validating the monotonicity assumptions requires ground-truth per-module performance, which, unfortunately, is not available for real-world tasks. However, we do observe that the convergence is not affected. As shown in Figure 9, the accuracy of LLMSELECTOR converges quickly as the number of API calls during training increases for several tasks. We have also added a discussion on the monotonicity assumptions to Appendix C.4.
>
> ***[Optimization overhead]***: We conduct additional experiments to study the optimization cost of applying LLMSELECTOR to a range of tasks. As shown in Figure 9, the overall accuracy converges quickly as the training budget increases. In fact, the number of model calls required to converge is modest even for compound AI systems with several modules. For example, 24 API calls/data point are sufficient for LLMSELECTOR to converge on Self-Refine (on the Word Sorting task). The detailed results and analysis can be found in Appendix C.2.3.

---

### Official Review · Reviewer_m2vk · 2025-11-01

**Soundness:** 3
**Presentation:** 4
**Contribution:** 3
**Rating:** 6
**Confidence:** 3

**Summary:**

This paper tackles the Model Selection Problem (MSP) in compound AI systems, where frameworks such as Self-Refine and Multiagent-Debate that combine multiple large language model (LLM) calls to solve complex tasks. The authors first formalize MSP and prove that finding the optimal model allocation is NP-hard. To address this challenge, they propose LLMSelector, a two-stage optimization framework that (1) learns each model’s module-wise performance using an LLM evaluator through a quality learning phase, and (2) performs module-wise ascent to efficiently identify near-optimal model allocations. Under mild assumptions, LLMSelector achieves linear computational complexity with respect to the number of models and modules. Experiments across diverse systems and tasks show that LLMSelector delivers substantial performance gains over fixed single-model baselines.

**Strengths:**

1. The paper systematically studies the model selection problem in compound AI systems, which is important and underexplored beyond traditional prompt or interaction optimization. Its efforts on handling the exponentially large search space of multi-module LLM pipelines is both novel and highly relevant for advancing automated AI pipeline optimization.
2. The proposed LLMSelector framework is theoretically grounded and achieves linear computational complexity under reasonable assumptions. Its design, combining quality learning with module-wise ascent, is effective and computationally efficient for compound systems.
3. The experiments, conducted across multiple tasks, datasets, and both frontier and general-purpose LLMs, demonstrate consistent and substantial performance improvements.
4. The paper is well-written and well-organized. The figures, examples, and case studies clearly convey the workflow, motivation, and contributions.

**Weaknesses:**

1. The efficiency and optimality proofs depend on strong monotonicity assumptions that may not hold in real-world settings. The paper should discuss sensitivity to assumption violations and expected performance impacts.
2. More ablation studies and comparisons under varied experimental setups are needed to clearly demonstrate advantages over existing compound system optimization methods.
3. The current framework is limited to pre-defined static architectures. It is interesting to explore dynamic or adaptive compound systems, where module structures or interactions evolve during inference.

**Questions:**

1. It is better to provide more implementation details on the quality learning phase.
2. It would be helpful to include a clearer analysis of the LLM call cost and computational overhead involved in the LLMSelector optimization process.

---

> ### Author Response · Authors · 2025-11-23
> **Thank you very much for supporting our paper!**
>
> Thank you very much for supporting our paper! We address your questions as follows.
>
> ***[Discussion on sensitivity to assumption violations]***: Thank you for the insightful suggestion. The monotonicity assumptions serve two purposes in our analysis section: (i) enabling clean forms of performance guarantees and (ii) formalizing intuitive properties observed on compound AI systems. The algorithm (Algorithm 1 on page 6), on the other hand, does not require these assumptions. Empirically, Algorithm 1 consistently leads to notable performance gains on real-world tasks when the monotonicity assumptions do not necessarily hold. For example, as shown in Figure 5 on page 8, applying LLMSelector to Self-Refine leads to 10%+ accuracy gains, even though the monotonicity assumptions may not hold for Self-Refine. This suggests that LLMSelector is robust to imperfect monotonicity and that the assumptions, while useful for analysis, are not critical for strong empirical performance. We have added the discussion to Appendix C.4.
>
> ***[Explore dynamic compound AI systems]***: We thank the reviewer for this insightful suggestion. The goal of this paper is to lay the foundation of LLM selection in compound AI systems, an under-explored area in compound system optimization. As a first step, we focus on static systems, simple, clean and widely-used scenarios where the technical depth and empirical benefits of LLM selection have already emerged. We totally agree with the reviewer that extending this to dynamic compound AI systems would be an exciting future direction. We have added a discussion on this to Appendix C.4 to reflect this.
>
>
> ***[Quality learning details]***: The key component in quality learning is the LLM evaluator, the details of which in our experiments are presented in Appendix C.2.2. In particular, we use GPT-5 as the LLM evaluator for experiments with frontier models, and Gemini 1.5 Pro for experiments with general-purpose models. The prompt and an ablation study on different LLM evaluators’ effects on the Locate-Solve system using general-purpose models are also included. For better coverage, we have conducted additional experiments to study the effects of LLM evaluators when applying LLMSelector to various systems. As shown in Figure 8 in the updated appendix, we observe that even weak LLM evaluators (such as GPT-5 Nano) enable LLMSelector to outperform vanilla compound systems, and powerful LLM evaluators often lead to fast convergence. We have updated Appendix C.2.2 to reflect the additional ablation and analysis.
>
>
> ***[Optimization overhead]***: We have conducted experiments to measure the optimization overhead for all systems studied in our paper. In particular, we study how the training budget (measured by the number of model calls per training sample) affects LLMSelector’s test accuracy on Self-Refine, Majority-Vote, Multiagent-Debate, and Locate-Solve. As shown in Figure 9 (see Appendix C.2.3 on page 21), larger budgets usually lead to better end-to-end performance. Consider LLMSELECTOR applied to the Self-Refine system on the Word Sorting task as an example. 16 model calls per training query lead to an accuracy of 61%, while 60 model calls give an accuracy of 65%.  Furthermore,  dozens of model calls (per training sample) are sufficient for convergence. For example, training LLMSELECTOR to reach the highest accuracy of Multiagent-Debate on the Health dataset only requires 40 model calls per training query. This suggests that the training overhead is tolerable when the compound system is expected to be used in large-scale applications. In addition, LLMSELECTOR allows users to specify the training budget, offering flexible choices especially for budget-limited users. The detailed results and analysis have been added to Appendix C.2.3.

---

### Author Response · Authors · 2025-11-23
**Thank you for the helpful feedback!**

We thank all reviewers for their helpful feedback! We have addressed their questions below and revised the paper (including the supplementary material) accordingly. All changes in the paper are highlighted in blue.

---

### Comment · Area_Chair_oMgN · 2025-11-25

Dear Reviewer,

Thank you for reviewing for ICLR. Since the discussion deadline is coming soon, could you please take a look at the author's rebuttal, respond to their comments, and update your rating as well? Thanks!

Best Regards

AC

---

### Meta-Review · Area_Chair_LFJG · 2026-01-07

**Summary:**

While the paper studies an interesting problem—model selection in compound AI systems—and presents a clear formulation with empirical results, the reviewers raise several substantive concerns that limit its suitability for acceptance. In particular, the theoretical guarantees rely on strong monotonicity assumptions that are unlikely to hold in practice, weakening the relevance of the analysis. Several reviewers also question the level of algorithmic novelty, noting that the approach largely combines existing techniques such as LLM-as-a-judge with greedy or coordinate ascent optimization.

From an empirical standpoint, although large gains are observed on synthetic or highly modular tasks, improvements on standard benchmarks are more modest, raising concerns about generality and real-world impact. Reviewers further highlight limited comparisons with state-of-the-art multi-agent or adaptive systems, insufficient cost analysis for larger settings, and restricted evaluation scope given the reliance on closed-source models.

Overall, despite being clearly written and addressing a relevant topic, the paper does not yet provide sufficiently strong theoretical grounding, novelty, or empirical breadth to meet the acceptance bar.

**Reviewer Concerns:**

Addressed by the rebuttal:
	•	The authors provided additional implementation details for the quality learning stage, including the LLM evaluator choice, prompts, and sensitivity analyses.
	•	Additional ablations clarified the role of the quality learning stage and showed faster convergence compared to random initialization.
	•	The authors expanded the discussion on optimization cost and provided empirical evidence that the method converges with a moderate number of API calls.
	•	Clarifications were added regarding dataset construction (e.g., TableArithmetic) and proof sketches previously deferred to the appendix.

Still outstanding:
	•	The core theoretical guarantees continue to rely on strong monotonicity assumptions that are unlikely to hold in realistic compound AI systems, limiting the practical relevance of the analysis.
	•	Concerns about algorithmic novelty remain, as the method primarily combines known components (LLM-as-a-judge and greedy/module-wise optimization) without a clearly new algorithmic contribution.
	•	The empirical evaluation still lacks comparisons against stronger or more recent state-of-the-art multi-agent or adaptive systems, making it difficult to assess relative advantages.
	•	Performance gains on standard, real-world benchmarks remain modest compared to those on synthetic tasks, leaving questions about generality and impact.
	•	Scalability to settings with many available models or dynamically structured systems remains insufficiently validated.

**Reviewer Scores:**

•	Reviewer m2vk (Score: 6)
Likely unchanged. While generally supportive, this reviewer expressed reservations about the strength of the assumptions and scope of the framework, which are not fully resolved.
•	Reviewer ujw1 (Score: 4)
Likely unchanged (4). Core concerns about novelty, realism of assumptions, and limited real-world impact remain largely unaddressed despite added ablations.
•	Reviewer A9Bg (Score: 4)
Likely unchanged (4). Concerns regarding lack of SOTA comparisons, cost, and reliance on closed-source models persist.
•	Reviewer T1GQ (Score: 6)
Likely unchanged. Although generally positive, this reviewer raised multiple concerns about applicability, scalability, and clarity that were only partially addressed.

---

### Decision · Program_Chairs · 2026-01-26

Reject